# Synergistic use of Sentinel-2 and UAV-derived data for Plant Fractional Cover distribution mapping of coastal meadows with Digital Elevation Models.

Ricardo Martínez Prentice[1], Miguel Villoslada[2,1], Raymond D. Ward[1,3], Thaisa F. Bergamo[1,2], Chris B. Joyce[1,4], and Kalev Sepp[1]

[1]Institute of Agriculture and Environmental Sciences, Estonian University of Life Sciences, Kreutzwaldi 5, EE-51006 Tartu, Estonia.
[2]Department of Geographical and Historical Studies, University of Eastern Finland, P.O. Box 111, 80101 Joensuu, Finland.
[3]School of Geography, Queen Mary University of London, London E1 4NS, UK.
[4]JBA Consulting, Haywards Heath, East Sussex, UK.

**Correspondence:** Ricardo Martínez Prentice (ricardo@emu.ee)

**Abstract.** Coastal wetlands provide a range of ecosystem services, yet are currently under threat from global change impacts. Thus, their monitoring and assessment is vital for evaluating their status, extent and distribution. Remote sensing provides an excellent tool for evaluating coastal ecosystems, whether with small scale studies using drones or national/regional/global scale studies using satellite derived data. This study used a fine-scale plant community classification of coastal meadows in Estonia derived from a multispectral camera on board Unoccupied Aerial Vehicles (UAV) to calculate the Plant Fractional Cover (PFC) in Sentinel-2 MultiSpectral Instrument sensor (MSI) grids. A Random Forest (RF) algorithm was trained and tested with vegetation indices (VIs) calculated from the spectral bands extracted from the MSI sensor to predict the PFC. Additional RF models were trained and tested after adding a Digital Elevation Model (DEM). After comparing the models, results show that using DEM with VIs can increase the prediction accuracy of PFC up to two times ($R^2$ 58-70%). This suggests the use of ancillary data such as DEM to improve the prediction of empirical machine learning models, providing an appropriate approach to upscale local studies to wider areas for management and conservation purposes.

## 1 Introduction

Globally, coastal wetlands provide a wide range of ecosystem services including flood and wave attenuation (Möller et al., 2014), estuarine filtration (Celis-Hernandez et al., 2022; de Lacerda et al., 2022), biodiversity maintenance (Sutton-Grier and Sandifer, 2019), and climate mitigation through carbon sequestration and storage (Martinetto et al., 2023; Maxwell et al., 2023). However, these ecosystems are under threat from a range of stressors including climate change (Ward et al., 2016b; Mafi-Gholami et al., 2019; Ward, 2020), pollution (Celis-Hernandez et al., 2020; Li et al., 2021) and direct losses through conversion to other land uses (de Lacerda et al., 2021).

Boreal Baltic coastal meadows support a diverse mosaic of plant communities (Burnside et al., 2007) each supporting a range of ecosystem services (Villoslada Peciña et al., 2019) and biodiversity, largely controlled by the microtopography (Ward

et al., 2015), providing a range of habitats for groups including arthropods, amphibians and waders (Rhymer et al., 2010; Rannap et al., 2016; Torma et al., 2018). Therefore, it is important to be able to map the extent and distribution of these plant communities (Ward et al., 2013; Villoslada et al., 2020).

Remote Sensing techniques are increasingly used to map the distribution of coastal meadow plant communities (Villoslada et al., 2020; Martínez Prentice et al., 2021) and to estimate biomass and sward structure using Unoccupied Aerial Vehicles (UAVs) (Villoslada Peciña et al., 2021). The very high spatial resolutions supplied by UAV-borne sensors also allow fine-grained ecosystem properties to be unveiled, which otherwise remain concealed under the coarse spatial resolution of satellites, such as plant fractional cover, soil organic carbon, or aboveground biomass (Heil et al., 2022). In addition, near-real-time monitoring routines and the avoidance of the effect of clouds are among the advantages of UAVs over satellite sensors (Colomina and Molina, 2014; Díaz-Delgado et al., 2019). Conversely, the disadvantages are not only their limited coverage and battery capacity but also the legislation restrictions and their dependency on the weather conditions, as well as the requirement to be in the field (Cracknell, 2017; Emilien et al., 2021).

On the other hand, Earth Observation satellites capture images with large swaths and a high temporal resolution, which allows the consistent study of large extents of ecosystems over multiple years. In the last decade, the idea of combining the high spatial resolution derived from UAVs with the large swath and regular revisit times of satellites has gained momentum. Some studies have successfully addressed the potential upscaling of UAV multispectral images to satellite image resolutions in order to address wetland biophysical variables at multiple scales (Laliberte et al., 2011; Díaz-Delgado et al., 2019) with UAV as a support for ground-truth observations. The accurate geometrical and radiometrical overlapping allows UAV imagery values to be aggregated into satellite pixel grids (Padró et al., 2018).

Remotely sensed data in combination with artificial intelligence are essential to supply comprehensive assessments of these shifts (Knight et al., 2006; Adam et al., 2010; Pettorelli et al., 2014), playing a key role in wetland mapping, ecosystem monitoring and trend detection, overcoming some of the difficulties of local wetland surveys with traditional in-situ field methodologies over large area extents, remoteness and inaccessibility (Mahdianpari et al., 2020). As the growing impacts of land-use intensification and climate changes become more conspicuous and widespread (Findell et al., 2017), local-scale field survey methods may not adequately reveal plant community shifts in a spatially-explicit manner to study spatio-temporal patterns in plant community distribution, environmental monitoring, and biodiversity conservation.

Modelling plant community coverage with remote sensing data is one of the main goals in ecological assessments and monitoring (Corbane et al., 2015). Combining remotely sensed data with Machine Learning (ML) algorithms shows robust performance due to their ability to deal with non-parametric distribution of ground-truth data as well as the multicollinearity of variables (Rodriguez-Galiano et al., 2012; E Thessen, 2016; Maurya et al., 2021). ML-based models are used to predict the presence of vegetation using indices as the input for different algorithms (Maurya et al., 2021), and Random Forest (RF) has been shown to be an accurate algorithm to predict Plant Fractional Cover (PFC) over large areas (Zhang et al., 2019; De Simone et al., 2021; Yang et al., 2020). Moreover, ancillary data such as Digital Elevation Models (DEM) interpolated from Light Detection and Ranging (LiDAR) point clouds have been successfully used for mapping plant communities in coastal

wetlands (Ward et al., 2013) together with multispectral data from remote sensing platforms. This combination provides an enhancement on the detection of new plant distribution patterns (Okolie and Smit, 2022).

The present study compared two PFC models of five plant communities in Estonian coastal meadows from Vegetation Indices (VIs) calculated with Sentinel-2 MultiSpectral Instrument (MSI) sensor and ancillary data from a DEM. High-resolution UAV imagery was used as the reference for PFC within the spatial resolution of a Sentinel-2 image. The main objectives were to: (1) quantify the relationship between UAV imagery and MSI imagery values; (2) predict and test ML models to predict individual PFC per plant community with VIs derived from Sentinel-2 spectral values; (3) build and test the performance of the models by adding DEM data to the VIs.

To improve the article's readability, we have included a list of abbreviations in Table 1.

**Table 1.** List of abbreviation and acronyms used in the paper

| Abbreviation | Definition | Abbreviation | Definition |
|---|---|---|---|
| UAV | Unmaned Aerial Vehicle | MF | Maximum Features |
| MSI | MultiSpectral Instrument | RMSE | Root Mean Squared Error |
| PS | Parrot Sequoia | MBE | Mean Biased Error |
| VI | Vegetation Index | LS | Lower Shore |
| PFC | Plant Fractional Cover | OP | Open Pioneer |
| DEM | Digital Elevation Model | US | Upper Shore |
| dGPS | Differential Global Positioning System | TG | Tall Grassland |
| DI | Initial Dataframe | RS | Reed Swamp |
| ID | Unique Identifier | KUD | Kudani |
| DF0 | Sampled Data Frame | TAN | Tahu North |
| DF1 | Data Frame 1 | TAS | Tahu South |
| DF2 | Data Frame 2 | RAL | Rälby |
| RF | Random Forest | MAT | Matsalu |
| N | Number of Estimators | RMP | Rumpo |

## 2 Materials and Methods

### 2.1 Study areas

Six coastal meadow study sites located in protected areas on the west coast of Estonia were selected for this study. Kudani (KUD), Tahu North (TAN) and Tahu South (TAS) belong to the Silma Nature Reserve; Rälby (RAL) and Rumpo (RMP), to the Vormsi Landscape Protection Area; and Matsalu (MAT), to the Matsalu National Park (Figure 1). These landscapes are characterized by coastal meadows extended over a gradual transition from the sea to terrestrial ecosystems with a low variation

of topography, typically 0 to 2 metres above mean sea level (Ward et al., 2016a). Sites were chosen based on their near-continuous management history, high conservation value for wading birds, and presence of endangered plant species (Rannap et al., 2004; Berg et al., 2012).

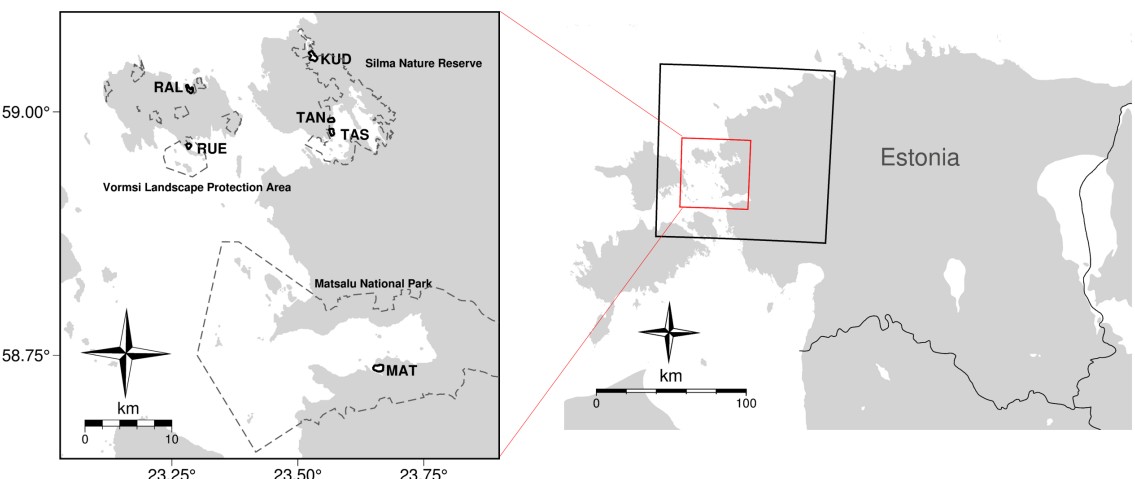

**Figure 1.** Location of the study sites in the western coast of Estonia. Matsalu (MAT), Tahu South (TAS), Tahu North (TAN), Kudani (KUD), Rälby (RAL) and Rumpo (RMP). The black square shows the Sentinel-2 tile footprint, whereas the extent of all the study areas is in red.

The plant communities under study are typical of Estonian coastal meadows and have been previously grouped following a phytosociological classification by Burnside et al (2007): Lower Shore (LS), Open pioneer (OP), Upper shore (US), Tall grass-

land (TG), and Reed Swamp (RS). This classification has been used in various studies in these coastal meadows and the plant communities have proven to be differentiable from high-resolution images (ca. 10 centimetres per pixel) (Ward et al., 2013; Villoslada et al., 2020; Martínez Prentice et al., 2021). The distribution of plant communities shows site-specific patterns due to local variations in the inundation levels and flood frequencies (Rivis et al., 2016), sediment accretion, microtopography (Ward et al., 2016a), and grazing regimes (Berg et al., 2012). Figure A1 shows the plant community distribution in relation to eleva-

tion in a boxplot. Floods depend mostly on the meteorological conditions across the North Atlantic and Fennoscandia (Kont et al., 2003) and the maximum level is reached in April after the snow melts. This is followed by the growing season, which is characterized by the maximum plant activity occurring from May until September. During this period, plant communities rely on mean temperatures above 10 °C (Paal, 1998).

Table 2 provides a distribution description of each plant community in this study.

**Table 2.** Summary of the description of plant communities in this study (Berg, 2008; Ward, 2012). Lower Shore (LS), Open Pioneer (OP), Upper Shore (US), Tall Grassland (TG) and Reed Swamp (RS).

| Plant Community | Description |
| --- | --- |
| Lower Shore (LS) | This community has adapted to significant variations in hydrological conditions, which result in the accumulation of litter and waterlogged soil, leading to considerable salinity concentrations. |
| Open Pioneer (OP) | This community of halophytic species is located in low-lying areas subject to prolonged inundation during the growing season, with its distribution primarily influenced by salinity. These specific locations exhibit the highest proportion of bare ground cover and highest salinity levels. |
| Upper Shore (US) | This community is established in higher elevations, characterized by less frequent and shorter floods. US is relatively more species rich and productive than LS. |
| Tall Grassland (TG) | This community is located on the highest elevations within the coastal meadows. Flooding is less pronounced and frequent, and vegetation is dense and very species-rich. |
| Reed Swamp (RS) | This community consists of extensive reedbeds along the coastline, which are influenced by more frequent inundations of brackish water. |

## 2.2 UAV classification data

A classification of the plant communities from high-resolution images (Martínez Prentice et al., 2021) was used as training/-validation. A multispectral Parrot Sequoia (PS) camera was carried on board of an eBee fixed-wing drone controlled remotely with the software SenseFly eMotion (Parrot S.A. Paris) over the six study areas at an altitude of 120 m to obtain Ground Sample Distance of 10 cm. The UAV flights were conducted during the dates corresponding to the growing season, carefully chosen to minimize the impact of inundation effects (Table 3). Images were radiometrically corrected with Airinov calibration panels and a sunshine sensor to produce multi band orthoimages merged in Pix4D v.4.3.31. VIs were calculated based on all the spectral bands (green, red, red edge and near infrared spectral bands) and used as input for two different workflows: a pixel-based classification, where the pixels were classified with a Random Forest and K-nearest neighbours' algorithms; and segmentation for an object-based classification with the same algorithms. The highest accuracy was achieved by a Random Forest pixel classification (accuracy and kappa greater than 90% and 0.85, respectively), calculated from a confusion matrix constructed using 140 vegetation survey quadrats as training samples (Figure A2), where all the species with coverage above 5% were recorded within the quadrats. A Sokkia GSR2700 ISX differential Global Positioning System (dGPS) was used to record the location and elevation of each plant community.

Plants in OP community were recorded as a result of the low cover of all species and predominance of bare ground. Not all plant communities were present in each site (Table A1). Further details of this methodology and results can be obtained in Martínez Prentice et al (2021).

## 2.3 Satellite imagery

Recent studies have shown that images taken by light-weight cameras in the visible and near infrared spectrum on board of UAV have a good correlation with satellite images, especially with MSI images of Sentinel-2 (Zabala, 2017; Zhu et al., 2021). Thus, one Sentinel-2 Level 2A image covering the six study areas (Figure 1) with the closest date to the drone flights was used (Table 3), with an estimated cloud cover of 19%. The tile number was T34VFL and its date, 24th of June of 2019. The level 2A was chosen because the orthorectified Bottom-of-Atmosphere reflectance values are comparable with PS reflectance (Fawcett et al., 2020). This image product is radiometrically corrected by the Payload Data Ground Segment with Sen2Cor algorithm (Main-Knorn et al., 2017) and available online via the Copernicus Scientific Data Hub tool (Copernicus Hub).

**Table 3.** Drone flight dates. Matsalu (MAT), Tahu South (TAS), Tahu North (TAN), Kudani (KUD), Rälby (RAL) and Rumpo (RMP).

| Study Area | Drone flight date |
|------------|-------------------|
| MAT | 29 June 2019 |
| TAS | 23 July 2019 |
| TAN | 30 June 2019 |
| KUD | 30 June 2019 |
| RAL | 04 July 2019 |
| RMP | 02 July 2019 |

Band 6 of MSI is in the Red-Edge region (Table 4) and contains valuable information of vegetation, avoiding background reflectance that affects wetlands especially (Turpie, 2013). Its spatial resolution is 20 metres per pixel. To use its reflectance values with the highest spatial resolution corresponding to the VNIR bands at 10 meters, an enhancement process based on a super-resolution method was applied, instead of using a panchromatic band to carry out a pan-sharpening since this band does not exist in MSI. The super-resolution algorithm (Brodu, 2017) is available in the SNAP software (ESA, 2014) and combines the geometric and radiometric information of target bands to increase the spatial resolution.

**Table 4.** Comparison of spectral resolution of bands in both sensors: MultiSpectral Instrument (MSI) on board of Sentinel-2 and Parrot Sequoia (PS) on board of eBee. First number is the central wavelength and the second one is the wavelength width. Units are in nanometers (nm).

| Band | MultiSpectral Instrument | Parrot Sequoia |
|---|---|---|
| Green | 559.8, 35 | 550, 40 |
| Red | 664.6, 30 | 660, 40 |
| Red Edge | 740.5, 14 | 735, 10 |
| Near Infrared | 832.8, 105 | 790, 40 |

## 2.4 Digital Elevation Models

DEMs constitute a powerful co-predictor in species distribution models, due to the prominent role of elevation in the distribution patterns of coastal plant communities (Ward et al., 2013). This holds especially true for coastal meadows, characterized by pronounced salinity and moisture gradients due to small variations of elevation, called microtopography (Ward et al., 2016a). Thus, a high-spatial resolution DEM was included in the models to test whether prediction accuracies improved. A LiDAR-derived DEM was downloaded from Eesti Maa-amet (Maa-amet geoportaal) with a spatial resolution of 1 m. The DEM was interpolated from a LiDAR point cloud of density of 2.1 points per square meter using a streaming triangulation (Isenburg et al., 2006). The vertical error was calculated with RMSE between in-situ elevation points of the DEM (Figure 2).

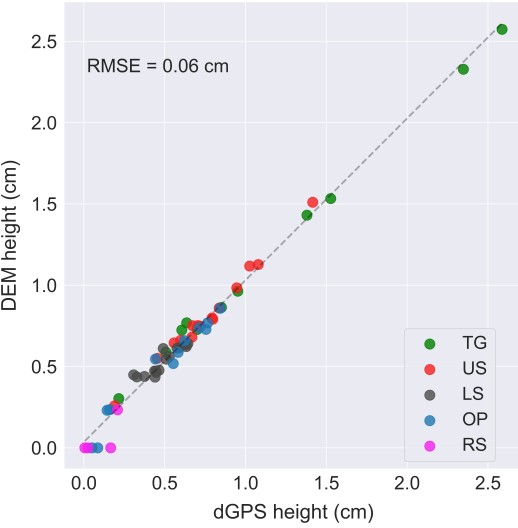

**Figure 2.** Vertical error between the measured heights with the differential Global Positioning System (dGPS), Sokkia GSR2700 ISX in each sampled plant community and the Digital Elevation Model (DEM). RMSE units are centimeters (cm). Lower Shore (LS), Open Pioneer (OP), Upper Shore (US), Tall Grass (TG), Reed Swamp (RS).

## 2.5 Image processing and upscaling

A key process required to perform upscaling of remote sensing images is the aggregation of pixel values from a high-resolution image to the geographically coincident pixels of coarser resolution image. Several studies have performed the aggregation process to a common geographical data frame in the form of a quasi-continuous grid, where all the spectral data is stored (Padró et al., 2018; Riihimäki et al., 2019; Mao et al., 2022; Bergamo et al., 2023). In the present study, the grid was constrained to the limits of each study area, avoiding those overlapping with the edges, excluding transitional areas that do not correspond to the extent of plant communities of interest and submerged areas. In total, 9766 MSI pixels cover the study areas (Figure A3).

A band-to-band comparison between the PS bands used for the final classification in Martínez Prentice et al. (2021) and MSI reflectance values was undertaken to assess the potential differences in both sensors caused by different temporal, spectral or spatial resolutions (Padró et al., 2018; Fernández-Guisuraga et al., 2018; Jiang et al., 2022; Isgró et al., 2022). To carry out this process, PS and MSI reflectance values were transferred into a polygon grid generated with the exact cell size as the MSI image pixels covering the study areas with an associated Unique Identifier (ID) for each row of the data frame (Figure 3). Level 2A MSI reflectance values were transferred to each cell of the polygon grid and the PS values were aggregated calculating the average mean (Figure 3). This approach generalizes the reflectance within a unit of grid, reducing noise from high-resolution images of PS and resulting in more predictable behavior (Blan and Butler, 1999). This aggregation criteria was also used to integrate the DEM values into the polygon grid (Figure 3).

The comparability and consistency of the spectral data from PS and MSI bands was analyzed by fitting the values in a linear model, calculating the coefficient of determination ($R^2$) and Root mean squared error (RMSE). The p-value showing the significance of the relation between PS and MSI.

The PFC was the response variable under assessment for each plant community. It was calculated by intersecting the UAV-derived classification maps (Martínez Prentice et al., 2021) within each polygon grid (MSI pixel) and applying equation 1. All the grids sum a total of 1 (100% PFC).

For all the operations, all the pixels completely covered by each grid were extracted.

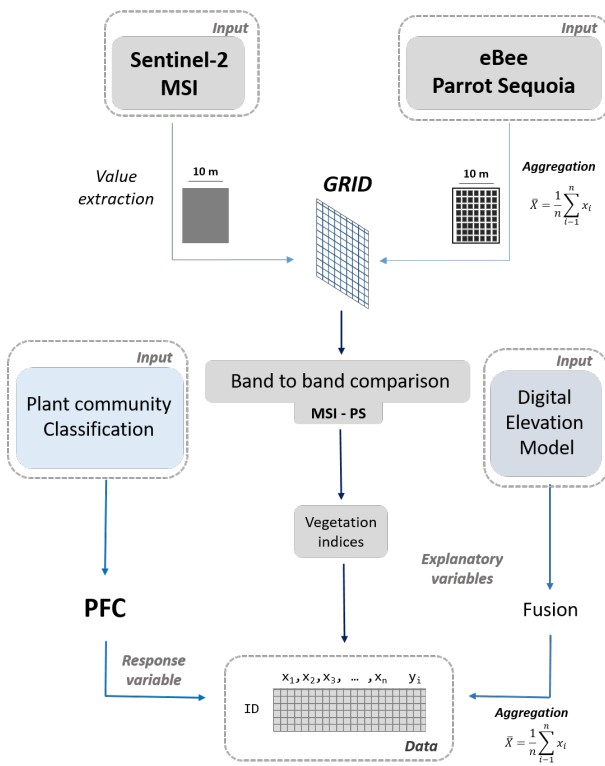

**Figure 3.** General workflow. The source data is marked as "Input" and the output Data Frame is DI. The final data frame contains the explanatory variables ($x_n$) and response variable ($y_i$) of Plant Fractional Cover (PFC), where i = Lower Shore (LS), Open Pioneer (OP), Upper Shore (US), Tall Grass (TG), Reed Swamp (RS).

PFC was calculated within each MSI pixel in the main data frame after an overlay process of the classification and MSI pixel extents. The Equation (1) was applied to each plant community of study.

$$PFC = \frac{Area\ of\ plant\ community\ within\ pixel\ extent}{Area\ of\ MSI\ pixel} \times 100 \tag{1}$$

All processes were carried out using the open source Python packages NumPy (Harris et al., 2020), GeoPandas (Jordahl et al., 2020) and rasterio (Gillies, 2013).

## 2.6    Vegetation indices

VIs are quantitative and dimensionless mathematical combinations of spectral bands, related to vegetation structural properties (Lima-Cueto et al., 2019). VIs have been used to monitor vegetation cover by the enhancement of spectral contrast between
photosynthetically active vegetation and other components (Andreatta et al., 2022). In this study, these band combinations

may unveil vegetation patterns related to different levels of flooding and phenological activity, even though the flight dates correspond to the growing season and water presence was at its lowest in the study areas (Table 3). Because of variations in the amount of bare ground within each plant community (Table 2), VIs are also used for their sensitiveness to this type of ground cover. In total, 14 VIs were calculated (Table 5) from all the MSI bands of this study. The red edge MSI band was included in the calculations of VIs because its reflectances show the highest photosynthetical activity and thus, better differentiation between plant communities (Schuster et al., 2012; Turpie, 2013). The indices in Table 5 were calculated by combining the features in the data frame (Figure 3) using the Pandas Python package (McKinney, 2010).

**Table 5.** List of fourteen vegetation indices used as explanatory variables in this study. G: Green band; R: Red band; Rre: Red Edge band; NIR: Near Infrarred band.

| Vegetation Index | Calculation | Reference |
|---|---|---|
| Normalized Difference Vegetation Index | $NDVI = \frac{NIR-R}{NIR+R}$ | Rouse et al. (1973) |
| Green Normalized Difference Vegetation Index | $GNDVI = \frac{NIR-G}{NIR+G}$ | Gitelson et al. (1996) |
| Chlorophyll Vegetation Index | $CVI = \frac{NIR \times R}{G^2}$ | Vincini et al. (2008) |
| Modified Simple Ratio (red edge) | $MSRred = \frac{(NIR/Rre)-1}{\sqrt{(NIR/Rre)+1}}$ | Wu et al. (2008) |
| Red edge triangular vegetation index (core) | $RTVI_{\text{core}} = 100 \times (NIR - Rre) - 10 \times (NIR - G)$ | Chen et al. (2010) |
| Canopy Chlorophyll Content Index | $CCCI = \frac{(NIR-Rre)/(NIR+Rre)}{(NIR-R)/(NIR+R)}$ | Barnes et al. (2000) |
| Chlorophyll Index (red edge) | $CI_{\text{re}} = \frac{NIR}{Rre} - 1$ | Gitelson et al. (2003) |
| Chlorophyll Index (green) | $CI_{\text{g}} = \frac{NIR}{G} - 1$ | Merzlyak et al. (2003) |
| Red edge normalized difference vegetation index | $NDVI_{\text{re}} = \frac{NIR - Rre}{NIR + Rre}$ | Gitelson and Merzlyak (1994) |
| Datt4 | $datt_4 = \frac{R}{(G \times Rre)}$ | Datt (1998) |
| Modified Green Red Vegetation Index | $MGRVI = \frac{(G^2-R^2)}{(G^2+R^2)}$ | Bendig et al. (2015) |
| Modified Soil Adjusted Vegetation Index | $MSAVI = \frac{2 \times NIR+1-\sqrt{(2 \times NIR+1)^2 - 8 \times (NIR-R)}}{2}$ | Qi et al. (1994) |
| Red Edge Ratio | $SR = \frac{NIR}{Rre}$ | Gitelson and Merzlyak (1994) |
| Green-red vegetation index | $GRVI = \frac{G-R}{G+R}$ | Chen et al. (2019) |

## 2.7 Machine Learning models

A ML algorithm was chosen to build each PFC model because this approach has been successfully used in various ecological
applications with Remote Sensing data (Olden et al., 2008; E Thessen, 2016). More specifically, the RF algorithm is widely
accepted because of its high performance in modelling species occurrence and distribution with remote sensing data without
making assumptions of data distribution (Evans et al., 2011; Shiferaw et al., 2019; Valavi et al., 2021). This algorithm was
chosen to build ten regression models.

To build the training and test samples, a stratified sampling from the Initial Data Frame (DI, Figure 3) was carried out. There
were two tails in the data distribution of PFC, showing a clear imbalance towards the lack of distribution or absence of plant
communities and a complete distribution of plant communities, typically happening in ecological data (Tang et al., 2023). To
account for it, the values of PFC were grouped in four bins created for this purpose ([0-25), [25-50), [50-75) and [75-100]).
Then, an under-sampling strategy was done by randomly reducing the number of values in each bin to match the number of
values in the minority bin (Table 6). This procedure balanced the distribution of PFC values across bins, avoiding potential
overfitting in the models that could result from learning more skewed bins while still capturing the entire dataset's variability.

**Table 6.** Balanced training dataset per plant community with the number of training rows considered in each bin and the proportion of all
the bins in relation to the number of all Sentinel-2 MultiSpectral Instrument (MSI) pixels (9766). Plant communities are: Lower Shore (LS),
Open Pioneer (OP), Upper Shore (US), Tall Grass (TG), Reed Swamp (RS).

| Plant | [0 - 25) | [25 - 50) | [50 - 75) | [75 - 100] | Total | Proportion (%) |
|-------|----------|-----------|-----------|------------|-------|----------------|
| LS    | 823      | 823       | 823       | 823        | 3292  | 34             |
| OP    | 178      | 178       | 178       | 178        | 712   | 7              |
| US    | 1169     | 1169      | 1169      | 1169       | 4676  | 48             |
| TG    | 711      | 711       | 711       | 711        | 2844  | 29             |
| RS    | 100      | 100       | 100       | 100        | 400   | 4              |

Two models were built for each plant community (LS, OP, US, TG and RS) from the Sampled Data Frame (DF0, Figure 4),
one trained with the list of 14 VIs as explanatory variables (Data Frame 1, DF1, Figure 4) and the other one adding the DEM
to the explanatory variables (Data Frame 2, DF2, Figure 4).

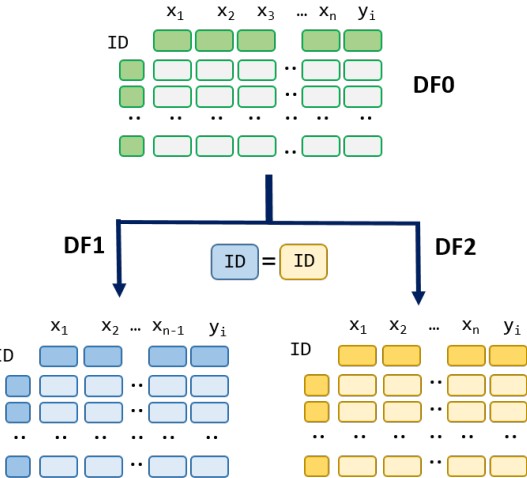

**Figure 4.** Diagram of the two training datasets used for the Random Forest (RF) models per plant community. DF0 is the sampled dataset after the under-sample strategy. DF1 has the same structure as DF0 except the DEM variable ($x_{n-1}$) and DF2 has all the explanatory variables ($x_n$). $y_i$ is the response variable (Plant Fractional Cover, PFC), where i = Lower Shore (LS), Open Pioneer (OP), Upper Shore (US), Tall Grass (TG), Reed Swamp (RS). DF1 and DF2 have the same samples (rows) matching the Unique Identifier (ID) column derived from DF0.

A fraction of 80% was used to train the RF regression models with DF1 and DF2 (Figure 4). A Grid Search Cross-Validation strategy was implemented to search for the best hyperparameters and tune a RF model (Figure 5). This method iterates through a grid of predefined hyperparameters and tests the results with a 10-fold cross validation (Figure 5). The hyperparameters used to carry out the grid search approach were the number of estimators (N) and maximum features (MF) used to find the best split to grow each tree in the forest. The standard parameters for RF (Probst et al., 2019) were not used in this study because preliminary results did not show acceptable $R^2$ and RMSE scores on the training dataset. The remaining 20% of the samples were used to test the trained model with the best hyperparameters. Using this approach, training and testing dependencies are removed, ensuring the robustness of the final model. In order to compare the RF models of plant communities trained with each dataset (DF1 and DF2), $R^2$, RMSE and Mean Bias Error (MBE) metrics were reported to quantify deviations between actual and predicted PFC. To account for the contribution of VIs and DEM in each model, the variable importance was also extracted. Variable importance is calculated during the training phase of the model, indicating the relative contribution of each single variable to each of the tree's total impurity reduction at each split of a node, meaning that it calculates the importance independently (Breiman, 2001). The importance ranges from 0 to 1 and it is calculated as the average of importance over all trees, indicating their relative contribution to the model accuracy. The models with the best scores were used to predict the relative PFC of each plant community over the whole polygon grid (Figure 5). Since the predicted values represent relative PFC within a MSI pixel, they were rescaled to a range of 0 to 100 and then validated with the test fraction again to assess the ability of RF regression models to predict absolute PFC. The RF models were programmed using the package scikit-learn in Python (Pedregosa et al., 2018).

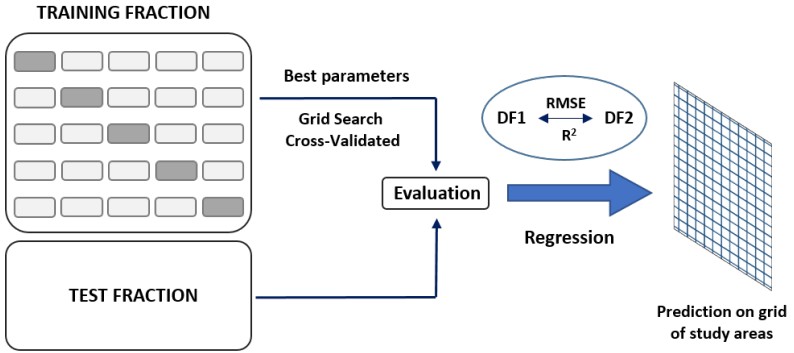

**Figure 5.** Machine learning algorithm training and testing process. A 10-fold cross validation on the training fraction (80% of the input dataset) was used to search for the best hyperparameters for the Random Forest (RF) model and the 20% for test fraction was used to test the trained model. The lowest Root Mean Square Error (RMSE) in the different RF models was used to predict the plant community distribution values on the polygon grid.

## 3 Results

### 3.1 Inter-sensor comparison

Figure 6 shows a quantitative comparison of spectral overlapping bands between MS and PS with $R^2$, RMSE and the significance level. Although the spectral resolution of PS and MSI sensors do not overlap completely (Table 4), the PS values aggregated by average into the MSI pixel show a significant positive correlation as well as low RMSE (Figure 6).

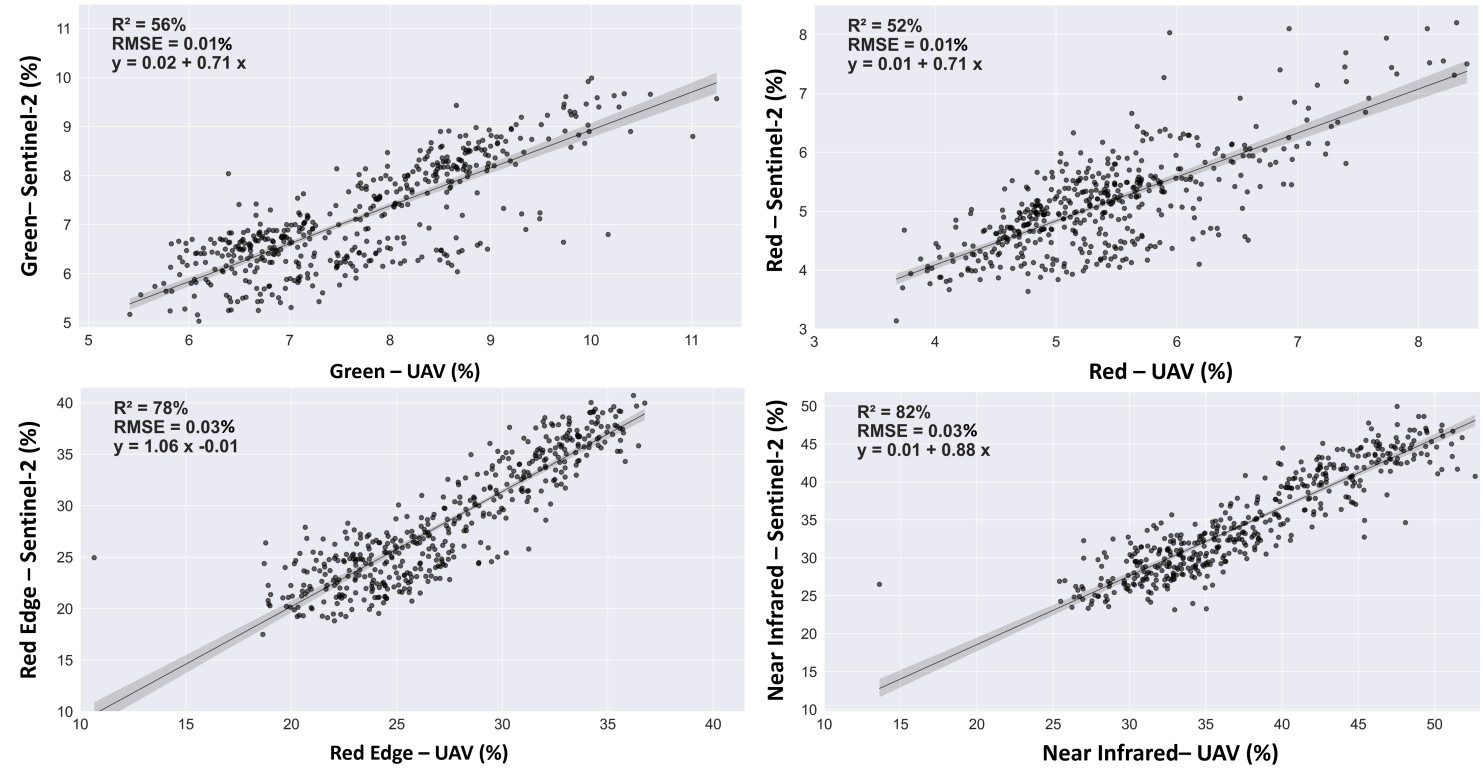

**Figure 6.** $R^2$ and RMSE obtained from the linear fitting between bands. X and y axes are in reflectance units (%) as well as RMSE. Correlations in all the cases are significant (p-value < 0.0001).

## 3.2 Random Forest regressions

The Grid Search Cross-Validation procedure enabled the selection of the best hyperparameters to build RF regression models with the lowest errors, leading to the identification of a minimum of 325 N. For models built on DF1, N was 500 except in RS (325) and OP (375) and 11 MF considered for the best split. Figure 7 shows the overall results of each RF regressor model with the best hyperparameters after the Grid Search in 10-fold cross validation. The models using only the VIs calculated from MSI bands (DF1) show a $R^2$ score under 57% and RMSE above 22% (units of PFC), resulting in a moderate to low prediction capability. Having 48% of the total samples to train, the RF model of the US community performed the worst with DF1, followed by TG and LS communities, which also had a greater percentage of samples to train the models (29 and 34%, respectively, Figure 7). The models built on DF2 required more N, from 400 to 500 and the same MF. These models showed a higher performance, where $R^2$ scores increased on average 20 units and RMSE decreased 5% on average (Figure 7). The best improvement of RF models is in US because the model trained and tested with DF2 increased its $R^2$ by 2.15 times (Figure 7). The highest $R^2$ was achieved by the RF model of TG, reaching 70% after training and testing with DF2. Its RMSE decreased

the most between models DF1 and DF2, from 27 to 19 % PFC. RF models trained and tested with DF1 and DF2 for LS, OP

and RS show the lowest differences of $R^2$ and RMSE despite having 34%, 7% and 4% of the samples for training and testing.

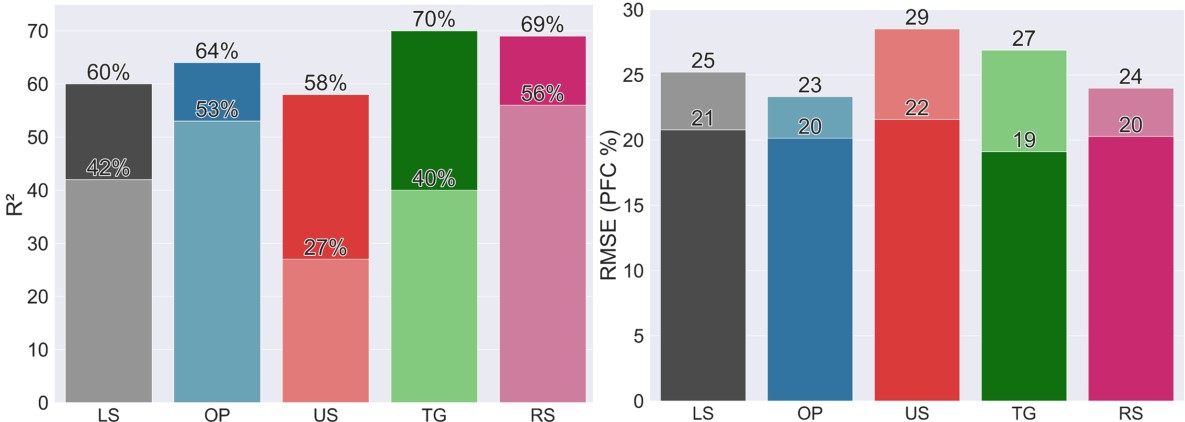

**Figure 7.** $R^2$ and Root Mean Squared Error (RMSE) retrieved by each Random Forest (RF) regressor on plant communities. Darker colours correspond to the model scores from Data Frame 2 (DF2) and lighter shades are model scores from Data Drame 1 (DF1). Plant communities are: Lower Shore (LS), Open Pioneer (OP), Upper Shore (US), Tall Grass (TG), Reed Swamp (RS).

The prediction errors in the RF models show a scattered distribution between the predicted and real PFC (Figures 8 and 9). In general, models tend to overestimate PFC below 50% of the real value and underestimate above it, according to the differences between the best fit line of the point distribution and the identity line (perfect prediction). This is more evident on the extreme values, around 0 and 100 PFC (Figures 8 and 9). These results improved in models on DF2, which showed the

best-fit line closer to the identity line than those on DF1. The MBE metric indicated that the models of LS, US and TG under and overestimate the same way, either trained with DF1 or DF2 (Figures 8 and 9). On the contrary, the models of OP and RS underestimated the predicted values mostly and did not show any improvement.

The average sum of relative PFC values predicted by RF models for each plant community within MSI pixels was 137%. After the validation of rescaled predicted PFC values from the best RF models, the RMSE and underestimations increased. The

results of these validations also show a steep decrease of $R^2$ (Figure 10). For these reasons, these models were not considered to map the PFC over the study areas.

Variable importance measured by the RF models to split the nodes did not show a common variable used in the models built with DF1 (Figure 11). Conversely, the models trained with DF2 show the DEM as a common important explanatory variable used to split the nodes, except for the model to predict PFC of OP (Figure 12).

The models trained with DF2 were used to map the distribution of relative PFC in the whole dataset (Figure 13) due to their lower RMSE.

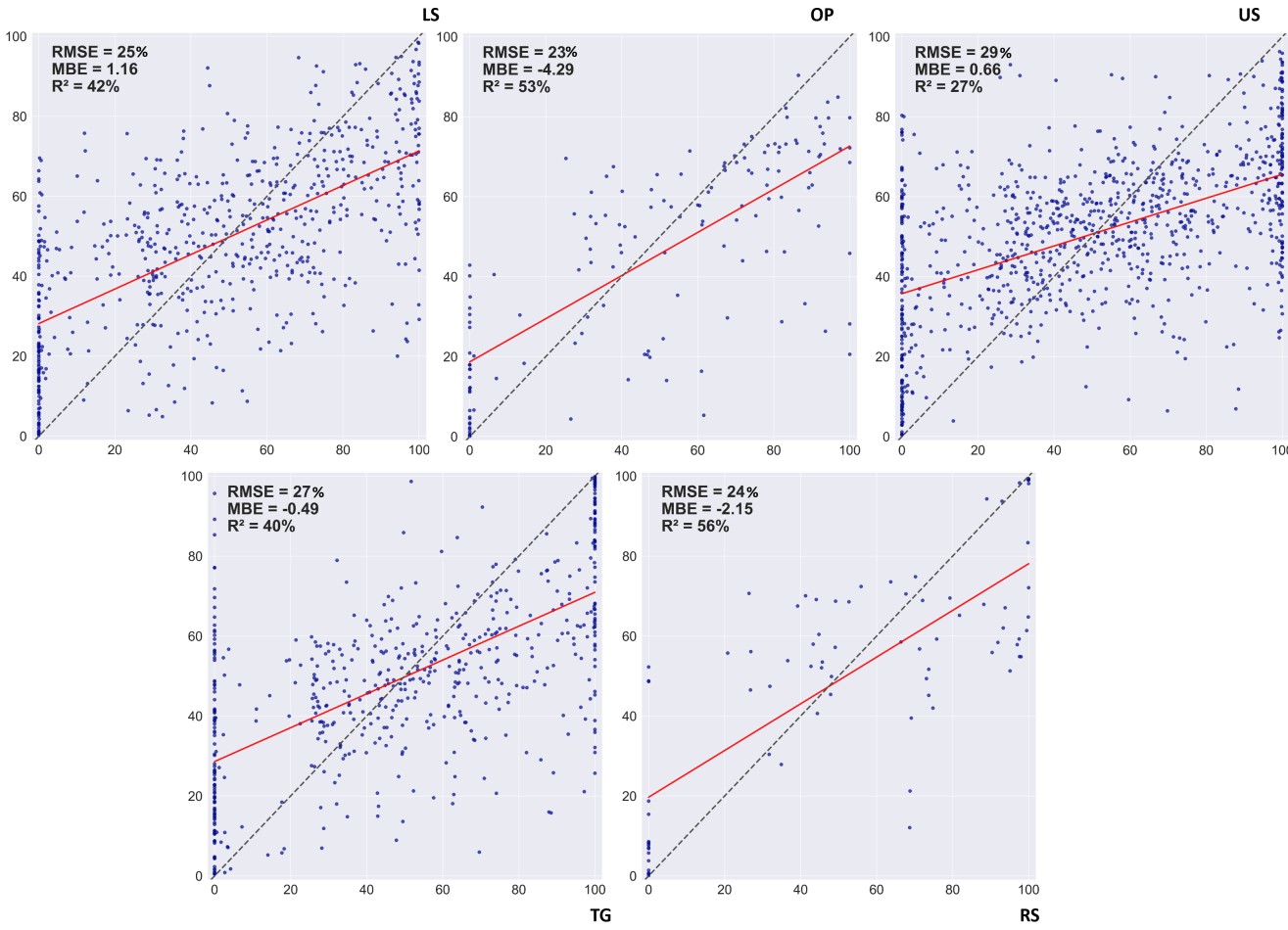

**Figure 8.** Prediction errors per plant community derived from Random Forest (RF) regressions with Data Frame 1 (DF1). On the x axis, actual values of Plant Fractional Cover (PFC, % ) and on the y axis, predicted values of PFC (%). Black dotted lines show the best fit estimated from the correlation between the predicted and measured value of the PFC (%). Red dotted lines represent the over or under estimation of the predictions with its quantification with Mean Biased Error (MBE) and Root Mean Squared Error (RMSE).Plant communities are: Lower Shore (LS), Open Pioneer (OP), Upper Shore (US), Tall Grass (TG), Reed Swamp (RS)

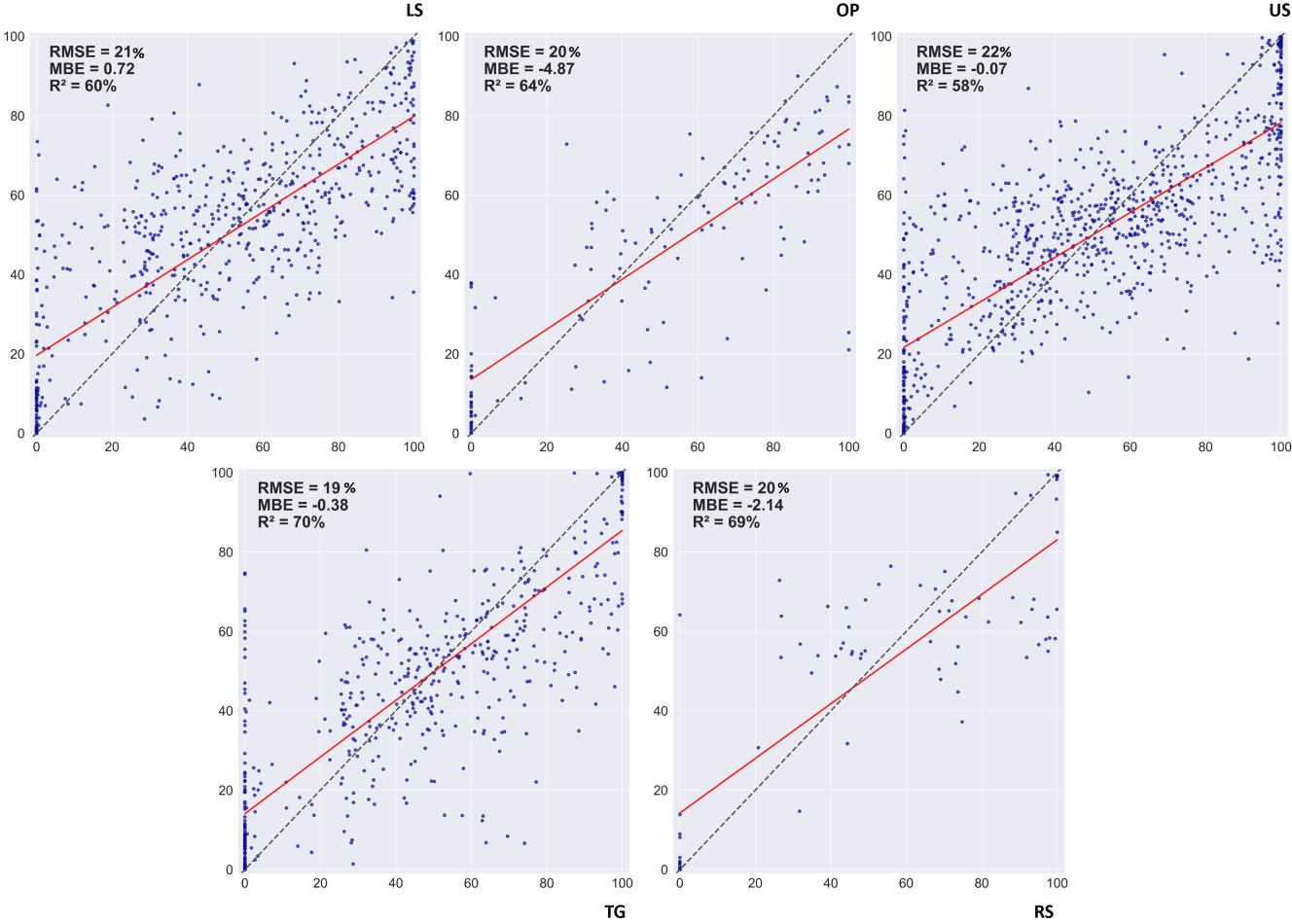

**Figure 9.** Prediction errors per plant community derived from Random Forest (RF) regressions with DF2. On the x axis, actual values of Plant Fractional Cover (PFC, % ) and on the y axis, predicted values of PFC (%). Black dotted lines show the best fit estimated from the correlation between the predicted and measured value of the PFC (%). Red dotted lines represent the over or under estimation of the predictions with its quantification with Mean Biased Error (MBE) and Root Mean Squared Error (RMSE).Plant communities are: Lower Shore (LS), Open Pioneer (OP), Upper Shore (US), Tall Grass (TG), Reed Swamp (RS)

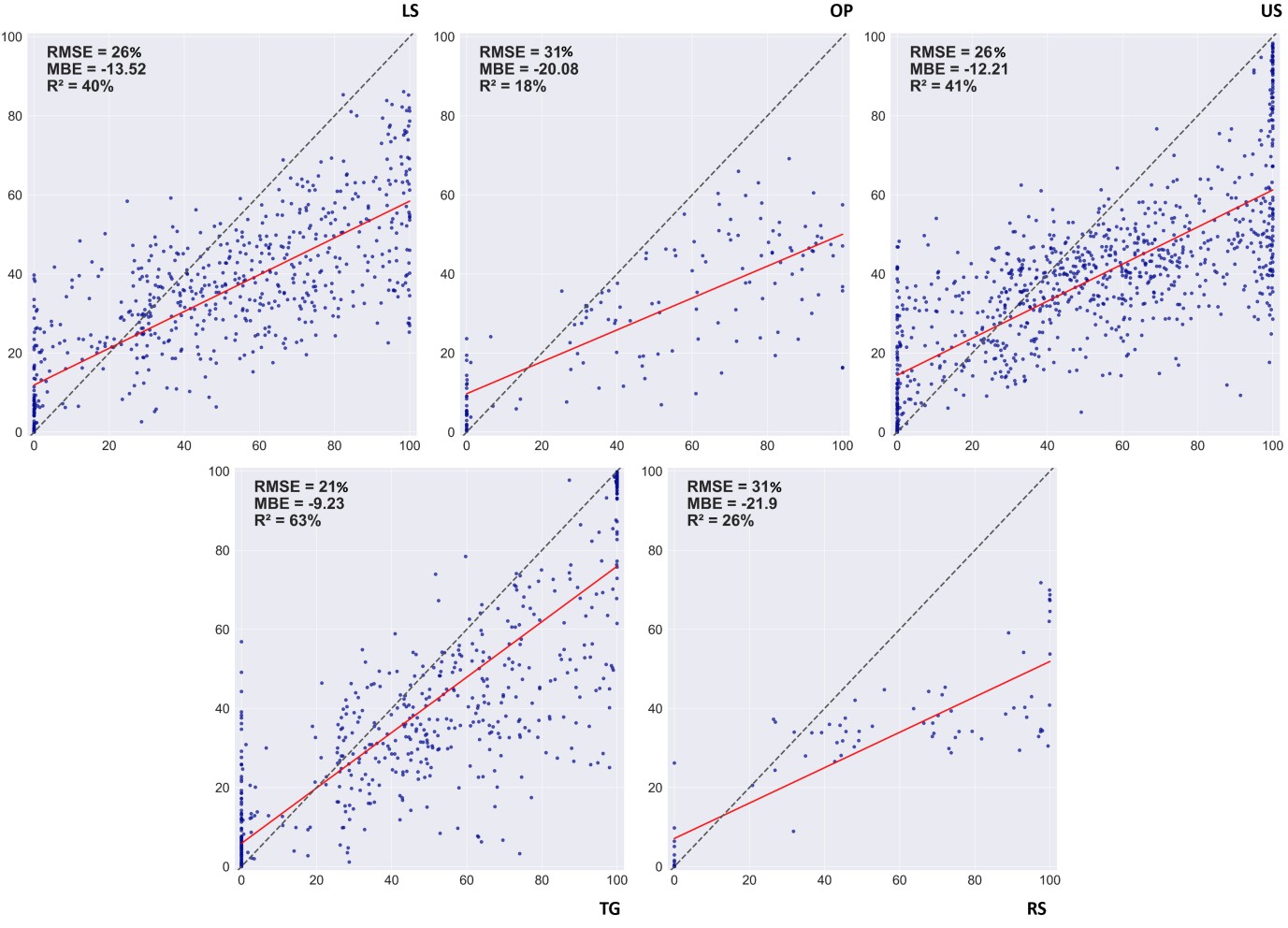

**Figure 10.** Prediction errors per plant community derived from Random Forest (RF) regressions with DF2 after rescaling prediction values. On the x axis, actual values of Plant Fractional Cover (PFC, % ) and on the y axis, predicted values of PFC (%). Black dotted lines show the best fit estimated from the correlation between the predicted and measured value of the PFC (%). Red dotted lines represent the over or under estimation of the predictions with its quantification with Mean Biased Error (MBE) and Root Mean Squared Error (RMSE).Plant communities are: Lower Shore (LS), Open Pioneer (OP), Upper Shore (US), Tall Grass (TG), Reed Swamp (RS)

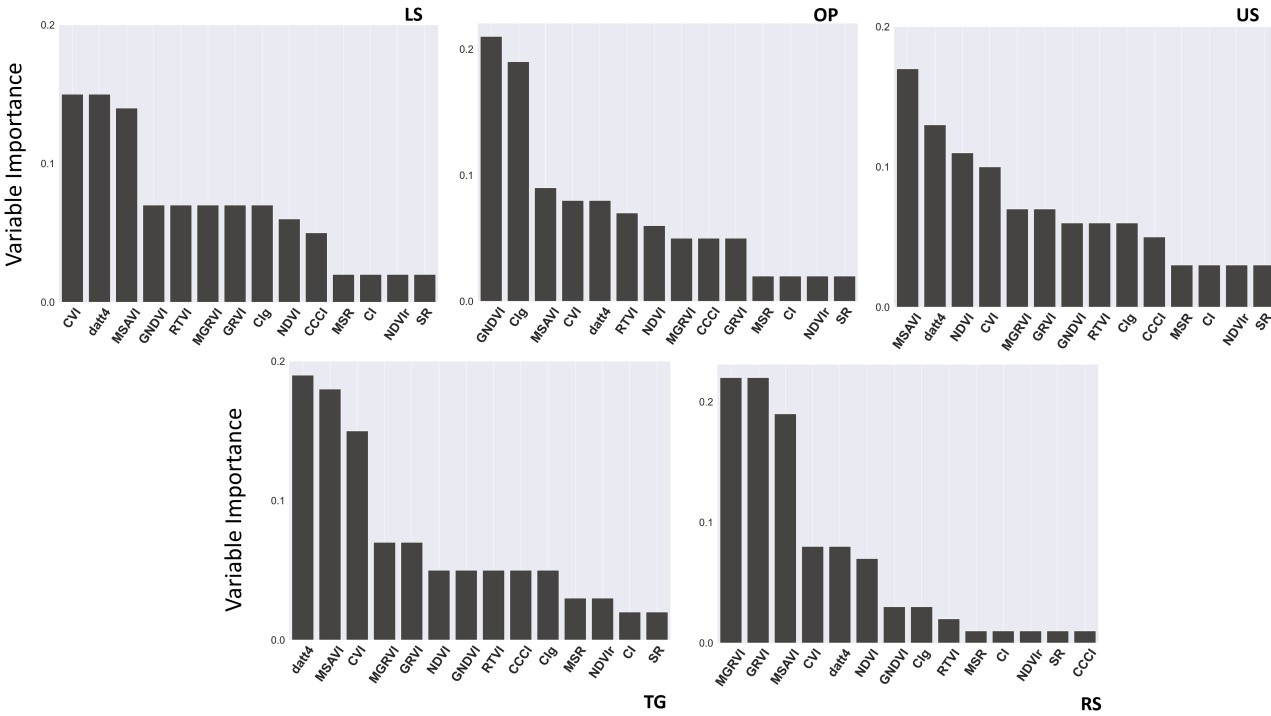

**Figure 11.** Variable importance retrieved by the Random Forest models derived from DF1.Plant communities are: Lower Shore (LS), Open Pioneer (OP), Upper Shore (US), Tall Grass (TG), Reed Swamp (RS)

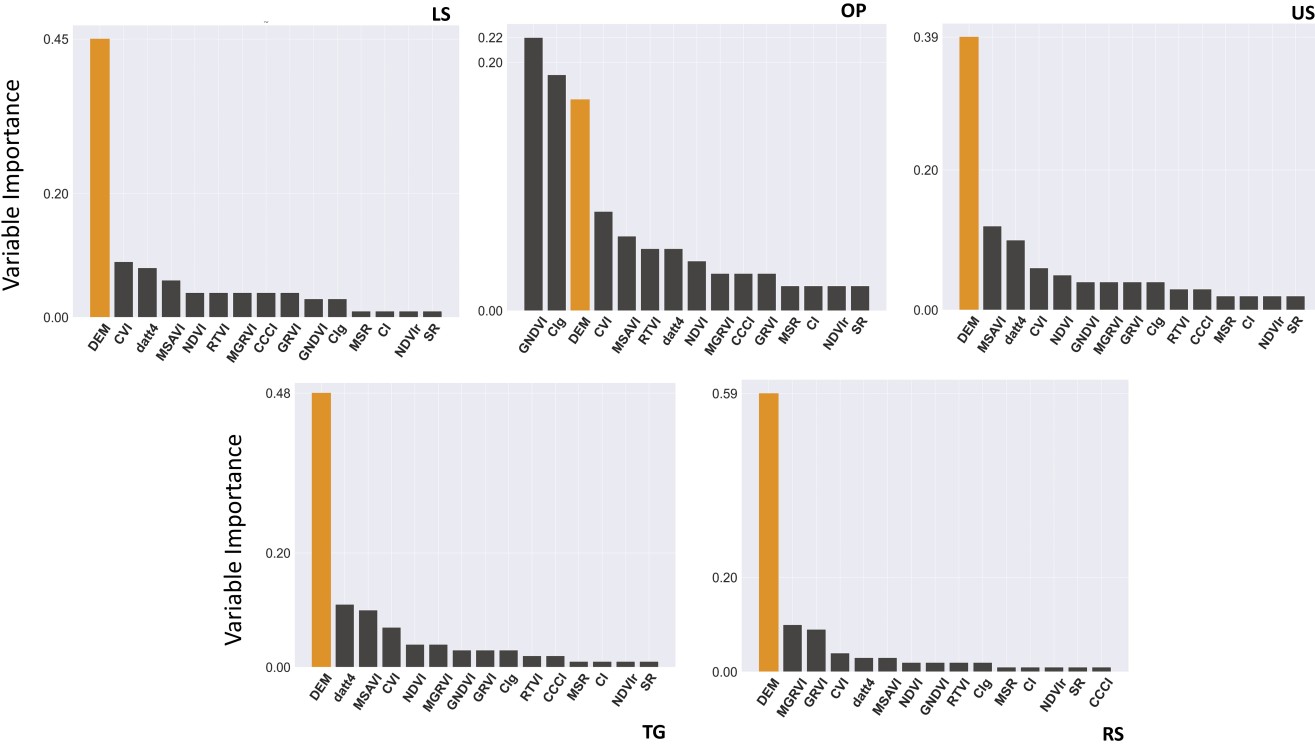

**Figure 12.** Variable importance retrieved by the Random Forest models derived from DF2, where the coloured bar represents the explanatory variable, Digital Elevation Model (DEM). Plant communities are: Lower Shore (LS), Open Pioneer (OP), Upper Shore (US), Tall Grass (TG), Reed Swamp (RS)

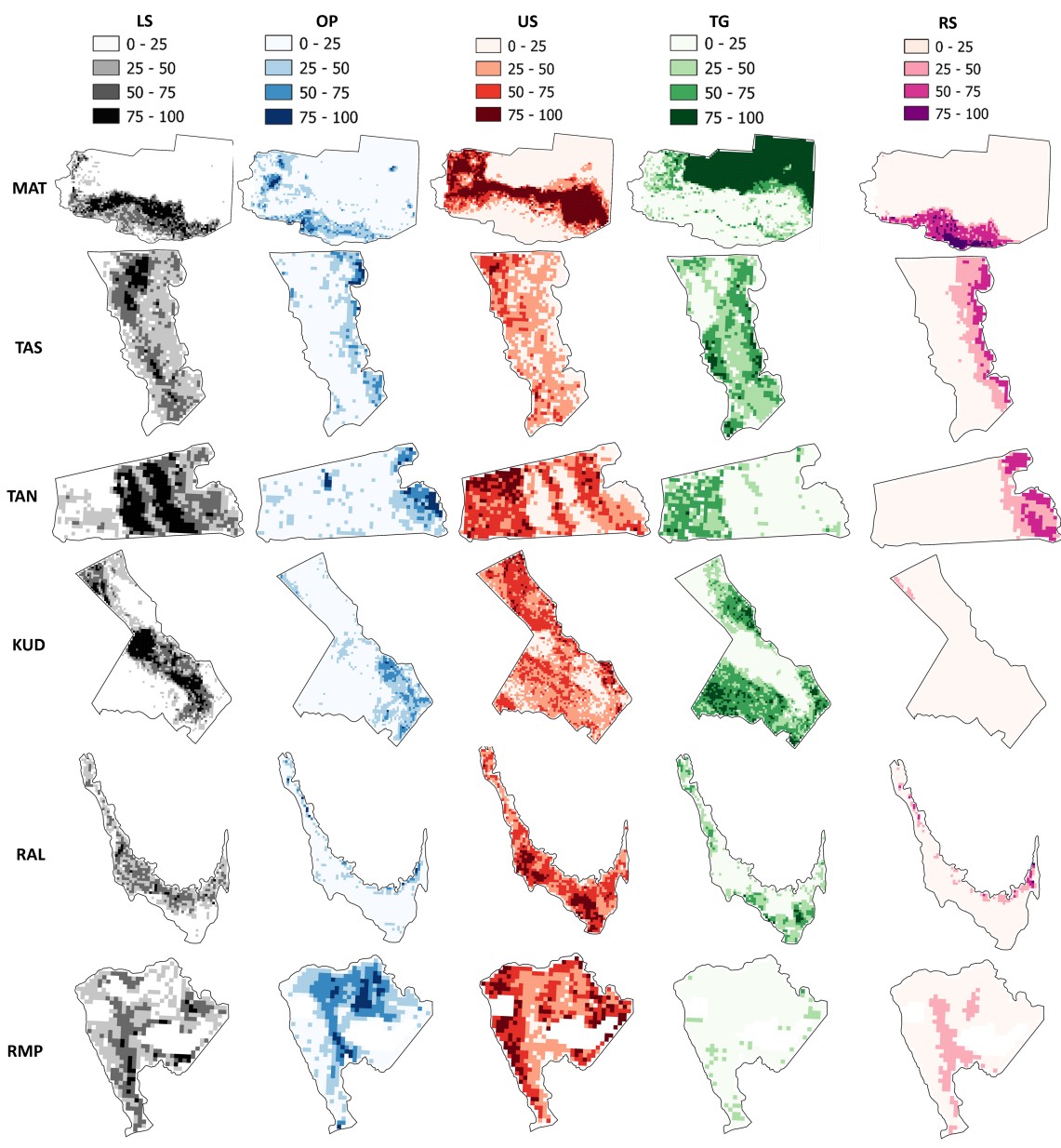

**Figure 13.** Maps of predicted Plant Fractional Cover (PFC ,% ) for each plant community within the study areas. Matsalu (MAT), Tahu South (TAS), Tahu North (TAN), Kudani (KUD), Rälby (RAL) and Rumpo (RMP). Plant communities are: Lower Shore (LS), Open Pioneer (OP), Upper Shore (US), Tall Grass (TG), Reed Swamp (RS)

## 4 Discussion

This study is one attempt to model the distribution of five coastal wetland plant communities belonging to the formal phytosociological categorization of Burnside et al., 2007, using open data from MSI sensor on board of Sentinel-2 and the official DEM of Estonia. A fine plant community classification within the study areas derived from the spectral bands of a PS camera (Martínez Prentice et al., 2021) was the reference to calculate the distribution of each plant community.

Firstly, the spatial aggregation by average mean of the PS images from 10 cm to 10 m gave coherent similarities with the values from MSI imagery at Level 2A after finding significant relationships between PS and MSI bands with a linear fit (Figure 6), having an average of 33% of unexplained variance in the relationships. The red and green bands in MSI data display weaker linear relationships when compared to the red edge and near-infrared bands. This discrepancy can be explained by the lower reflectance values in the visible spectrum within MSI, which might be influenced by the presence of a mixture of vegetation and water within the pixel. The PS data captured more reflectance of bare soil, contributing to higher reflectance values in red and green bands. Higher relationships are observed between red edge and near infrared bands from both sensors (Figure 6). This is because these bands capture strong reflectance signals from vegetation. Similar studies compared reflectance values of PS (Díaz-Delgado et al., 2019), or another sensor on board of UAV (Padró et al., 2018), with MSI images, finding good relationships.

Secondly, the RF algorithm used in this study assessed the accuracy of non-parametric based regression models to predict the distribution of plant communities, as commonly used in Earth Observation studies (Ferreira et al., 2022). In the literature, Random Forest is acknowledged as a robust algorithm for plant distribution at large scales (Maxwell et al., 2018; Butler and Sanderson, 2022), despite of the challenges in its interpretability (Simon et al., 2023). RF averages the predictions of individual trees, a process that contributes to the model's robustness and ability to generalize, however, underpredicts samples in either of the extremes, contributing to model's uncertainties (Kuhn and Johnson, 2013). In an effort to enhance the interpretability of the RF models, a balanced training datasets were employed to encompass the complete spectrum of PFC values. The undersampling approach resulted in improved models' performances, robustness, and generalizations, as evidenced by the minimal discrepancies between test and training scores observed during the Grid Search Cross Validation. A more in-depth examination of inherent model structures, specifically from hyperparameter tunning perspective considering splitting criterion or averaging functions on terminal nodes of each tree in the forest, should be considered. Moreover, constructing an exhaustive training dataset from field survey plots equal to the area of S2 pixels could reduce uncertainties, although it can be time-consuming due to logistic issues.

The predictions of PFC improved after the fusion of a high-resolution DEM (Figure 7). However, the mixture of reflectance signals included in one MSI pixel might have caused the deviations in the predictions (Figures 8 and 9). Overestimations of PFC in the models might be due to presence of patches where plant communities were mown or trampled, thus, retrieving VIs values near to bare ground values. Underestimation, on the other hand, is due to the mixture of reflectance responses from different plant communities within the same spatial distribution of MSI pixels. Martínez Prentice et al. (2021) suggest that the presence of disagreement areas is due to a mixture of radiances in transitional areas or ecotones. The effect of mixed pixels is

more significant in these types of transitional areas, as the sensor receives a wide range of reflectance signals within the extent of the pixel (Muukkonen and Heiskanen, 2007). The models of OP and RS with DF1 and DF2 underestimate PFC to a larger degree. The reflectance values retrieved from these plant communities are affected by a higher presence of water, reducing the values of VIs due to lower reflectance in the Near Infrared and Red Edge spectrum.

The decrease of RMSE in the models trained with DEM (Figure 7) suggests that plant communities follow the elevation pattern at broader scales corresponding to variations in microtopography in the study areas represented by the DEM. The variable importances derived from RF models using DF1 and DF2 as shown in Figures 11 and 12 respectively, show that the models consider the DEM to predict PFC as very important. Similar predictive efficiency by DEM are shown in those generated with the photogrammetric point cloud derived from UAV at a resolution below 10 cm (Villoslada et al., 2020), as
plant communities are strongly dependent on the variation of elevation represented with the DEM, in spite of being aggregated to 10 m. The only model that did not show the same importance of DEM was the RF model of OP, where the index GNDVI is the most important. OP is a type of plant community that is distributed over patches with a high proportion of moist and bare ground (Bergamo et al., 2022), where the visible part of the spectrum (Red and Green) retrieve greater reflectance values than the far visible part of the spectrum (Red Edge and Near Infrared). Ward et al. (2013) concluded that the presence of OP occurs
at similar elevation ranges than TG by predicting with a microtopography variable. Building models with the DEM alone do not reach the metrics as with the VIs, as these are indicators of different vegetation conditions, phenology and structure of plant communities. Despite the considereation of all variable importances, VIs are not additive representations of the spectral information because they are variables considereed in relative terms of importance by the models, therefore, they do not add together and explain PFC more than DEM.

The final plant community distribution maps match the common patterns recognised by expert knowledge in the study areas and the maps shown in Martínez Prentice et al. (2021) at a 10 cm spatial resolution (Figure 13). According to these criteria, the PFC maps in MAT are the most representative of PFC (Figure 13). Some plant communities that were not identified in most of the study areas (Table A1) are present with high PFC values. This is the case of RS because its distribution along the coastline shows similar low values of DEM and VIs due to the higher inundation levels of brackish water (Table 2). It is largely missing
in KUD mostly, as this study area is relatively far from to the coastline (Figure 1). The plant community of TG is overestimated in TAN and RMP areas where it was not identified, caused by similarities in VIs and DEM with other plant communities, which means similar biogeographical factors than in other study areas where it is actually present. The distribution of LS, OP and US are mostly correct according to the aforementioned criteria. OP presents a similar overestimation as RS of its PFC in RAL area only, for the same reason concerning RS.

While the fusion of DEM with VIs increases the accuracy of RF regression models in predicting relative PFC within MSI pixels, being a valuable quantification of individual plant community distribution from Sentinel-2 images (Figure 13), they are not suitable to predict the absolute PFC, as they yield higher estimation errors ( comparison of Figures 9 and 10). Due to the high deviations of predicted relative PFC (average of 137%), once the values are rescaled, the plant communities with low distribution become even lower, thus, giving underestimated predictions. Similar prediction errors after rescaling were
also noted in Yang et al. (2020) using a higher satellite spatial resolution. Our validation was done from a fine-scale plant

community classification, being a reliable resource of an exhaustive vegetation inventory because it reflects the small-scale variations and mixed plant communities typical in these coastal wetlands.

This study shows an acceptable empirical approximation to modelling the distribution of ecosystem service-providing units in large-scale images of coastal meadows of Estonia, this is, quantifying the prediction accuracy and error of PFC of small sized plant communities in heterogeneous ecosystems at a satellite spatial resolution. The main advantage of using publicly available data from the Sentinel-2 constellation is its short revisit time, provided by the two satellites, Sentinel-2 A and B.

The methodology provides a rapid assessment of plant communities in a coastal ecosystem vulnerable to climate and land use changes using different sources of remotely sensed data. Additionally, it is shown that it is possible to reduce time and costs associated with multiple UAV flights in different areas to cover large extents by the validation of large-scale monitoring studies with open source satellite data such as Sentinel-2 and high-resolution products derived from multispectral images taken from UAV.

Upscaling remotely sensed imagery from fine to coarse resolution is necessary, although challenging. Satellite imagery provides critical information of changes needed for improved environmental management and conservation decision making at large scales. Considering this, linking UAV to Earth Observation satellites offers the opportunity for multiscale studies of environmentally sensitive ecosystems such as coastal wetlands. Further work can consider using ancillary data as a co-predictor with the aggregated spectral data, such as temperature, pluviometry or distance of plant communities from the coast, to improve the prediction accuracies, as shown with DEM data in the present work. The supervised learning RF algorithm is one of the most robust ML algorithms used for ecosystem and species distribution modelling (Pichler and Hartig, 2023), however, other algorithms should be explored, such as the soft classification, successfully implemented in (Yang et al., 2020). Moreover, recent advances in Super-Resolution methodologies increase the spatial resolution of Sentinel-2 images by four times for all bands, at a maximum of 2.5m using Artificial Intelligence algorithms (Tarasiewicz et al., 2023). Finer scale analysis will be more suitable to study the heterogeneity of plant communities such as those in Boreal Baltic coastal meadows.

## 5   Conclusions

A multiscale synergy approach between UAV and Sentinel-2 MSI was undertaken in this study to model the PFC of five plant communities in coastal meadows. Good relationships existed between both sensors, which enabled PFC to be modelled using VIs, although the fusion of DEM improved the models from 1.2 to 2 times. From this research, future studies on coastal meadows using remote sensing from satellites should be focused on finding methods to achieve local calibration of the image based on values retrieved from UAV mounted multispectral cameras and thus, achieve stronger synergies between both sensors (Emilien et al, 2021). Due to the high repeat time and long duration of data collection, upscaling from UAV to satellite imagery provides an excellent resource for monitoring and assessment of the response of coastal ecosystems to loss and degradation as a result of climate change or other anthropogenic stressors. This will allow land users and managers to appropriately assess conservation priorities and implement and monitor responses.

# Appendix A

## A1 Additional figures and tables

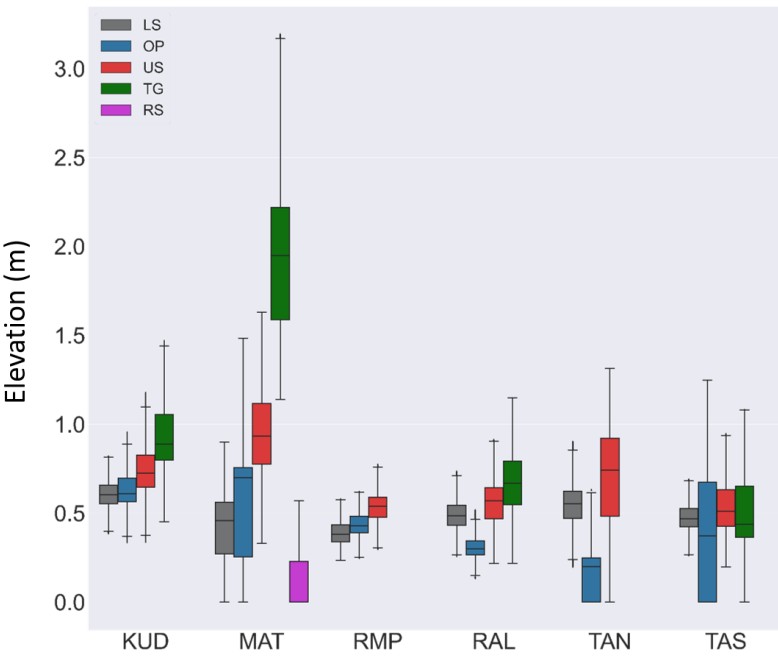

**Figure A1.** Boxplot of elevation ranges (m) per category of plant communities in each study area, showing the microtopography gradient within the established plant communities.Matsalu (MAT), Tahu South (TAS), Tahu North (TAN), Kudani (KUD), Rälby (RAL) and Rumpo (RMP). Plant communities are: Lower Shore (LS), Open Pioneer (OP), Upper Shore (US), Tall Grass (TG), Reed Swamp (RS)

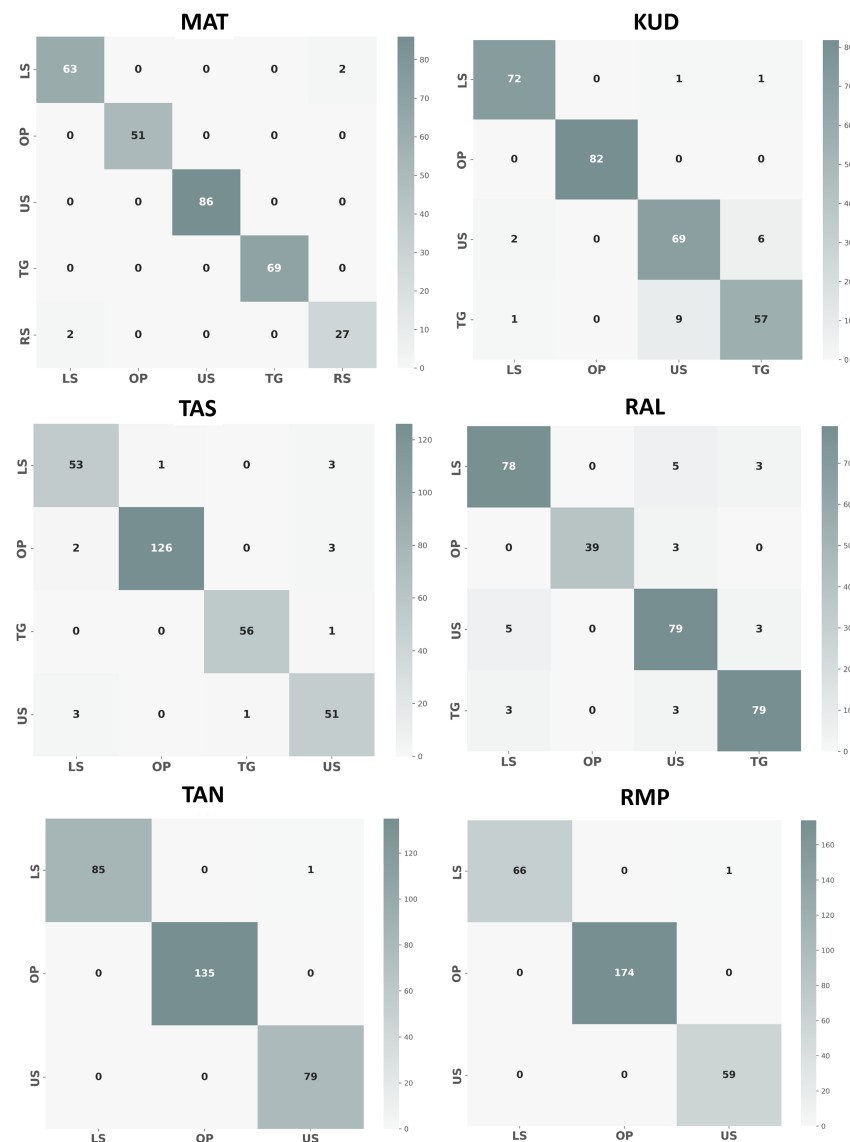

**Figure A2.** Confusion Matrix from the results of Random Forest pixel classification in Martinez Prentice et al., 2021. MAT (Matsalu), KUD (Kudani), TAS (Tahu South), RAL (Rälby), TAN (Tahu North) and Rumpo (RMP). Kappa values are MAT: 0.98, KUD: 0.92, TAS: 0.93, RAL: 0.89, TAN: 0.99 and RMP: 0.99. Each class of Predicted and Actual Plant Communities are LS (Lower Shore), OP (Open Pioneer), US (Upper Shore), TG (Tall Grassland) and RS (Reed Swamp). Numbers are the pixels classified in each quadrat

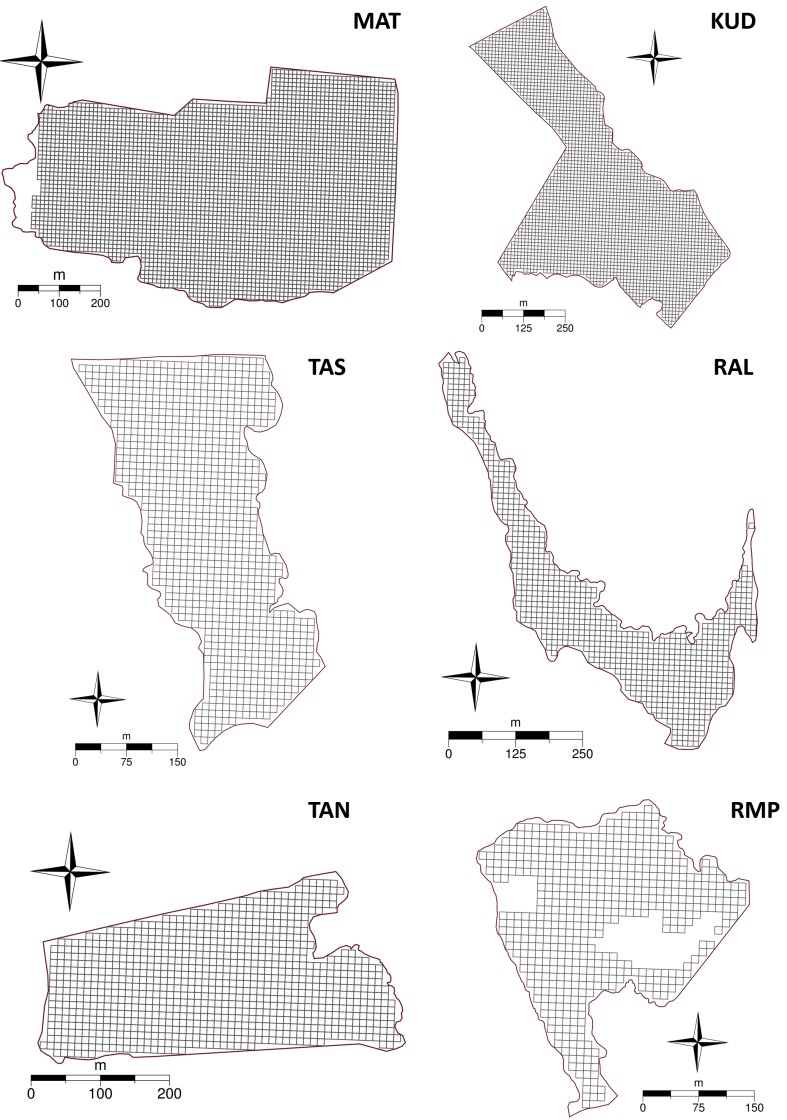

**Figure A3.** Polygon grids from Sentinel-2 MultiSpectral Instrument (MSI) pixels (9766) covering the six study areas. 1. Matsalu (MAT), 2. Tahu South (TAS), 3. Tahu North (TAN), 4. Kudani (KUD), 5. Rälby (RAL), 6. Rumpo (RMP)

**Table A1.** Plant communities sampled in each study area. Matsalu (MAT), Tahu South (TAS), Tahu North (TAN), Kudani (KUD), Rälby (RAL) and Rumpo (RMP); Lower Shore (LS), Open Pioneer (OP), Upper Shore (US), Tall Grass (TG), Reed Swamp (RS).

| Study Area | Plant Communities |
|:---:|:---:|
| MAT | LS, OP, US, TG, RS |
| TAS | LS, OP, TG, US |
| TAN | LS, OP, US |
| KUD | LS, OP, TG, US |
| RAL | LS, OP, TG, US |
| RMP | LS, OP, US |

*Author contributions.* Conceptualization, R.M.P., M.V.P. and R.D.W.; methodology, R.M.P., C.B.J., M.V.P., T.F.B. and R.D.W.; software, R.M.P. and M.V.P.; validation, R.M.P., M.V.P. and R.D.W.; formal analysis, R.M.P.; investigation, R.M.P., M.V.P., T.F.B., C.B.J. and R.D.W.; resources, R.D.W. and K.S.; data curation, R.M.P., M.V.P. and R.D.W.; writing—original draft preparation, R.M.P., M.V.P. and R.D.W.; writing—review and editing, R.M.P., M.V.P., R.D.W. and C.B.J.; visualization, R.M.P.; supervision, M.V.P., R.D.W. and K.S.; project administration, R.D.W. and K.S.; funding acquisition, R.D.W. and K.S. All authors have read and agreed to the published version of the manuscript.

*Competing interests.* The contact author has declared that none of the authors has any competing interests.

*Acknowledgements.* This research was funded by the Doctoral School of Earth Sciences and Ecology, financed by the European Union, European Regional Development Fund (Estonian University of Life Sciences ASTRA project "Value-chain based bio-economy".

Moreover, the authors would like to thank the anonymous reviewers for their valuable comments and critical questions, which helped to improve the quality of the present manuscript

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
