# Peer review of "Synergistic use of Sentinel-2 and UAV-derived data for Plant Fractional Cover distribution mapping of coastal meadows with Digital Elevation Models."

_Biogeosciences, 2023_

## Author Comment (AC1)

*First of all, we appreciate the comments provided. Your expert opinion is helping this paper to improve its robustness and to reach more readers.*

*I will answer your comments in the following lines. These answers are in 'Italics' :*

This paper talked about the upscaling of Drone image classification to open access satellite images, and applied to the coastal wetlands. I do think it is a good idea to overcome the disadvantages of low spatial resolution of open-access satellite images. It is a key step to achieve the continuous monitoring of coastal wetlands at a low cost.

However, I have a few concerns about this method. I suggest this manuscript needs to be major revised.

If I understand correctly, the author establishes an individual RF regression model for each community to each study site. My concern first concern is that have the authors united the plant fractional cover (PFC) of each landcover to make sure the sum value of the PFC of each community within each Sentinel pixel is equal to 1. If so, please highlight it or remind me where I can find it in the main text. If not, I suggest the authors to rescale the retrieved results and test the accuracy again. Some previous analyses have indicated that such rescaling can change the accuracy obviously (e.g., Immitzer et al., Remote Sensing of Environment, doi: 10.1016/j.rse.2017.09.031; Yang et al., Remote Sensing, doi: 10.3390/rs12193224).

*We created a dataset for training – validation and testing based on PC values of each plant community that sum 1 (this is, 100% cover) within each space unit of MSI pixel. We will write a clearer sentence in the main text, section 2.5, as follows: "PFC was extracted for each plant community within each MSI pixel. PFC sums 100% for each plant community".*

Here are some specific comments in the main text:

1. Do the authors think that tidal level would affect the accuracy of PFC estimation? Some previous works mentioned that tidal level can significantly affect the land cover classification (e.g., Kearney et al., 2009 Journal of Coastal Research, doi: 10.2112/08-1080.1). I suggest that the tidal level at the time of each drone image and Sentinal image acquisition should be reported

   *I will add the following clarification: "In Estonian coastal wetlands, the tidal variation is negligible (0.02 m) and the range of plant communities is maintained through low-intensity grazing (Ward et al. 2016)"*

2. The authors examine the accuracy of applications with and without elevation data provided by DEM. Can authors specify that did you use the same points to train and

test your two applications? I think it is important to show that adding DEM is useful. In addition, can the authors show the plots of DEM in the main text or supporting information?

*Definitely, we used the same points for both applications (models). Thus, we will add the following sentence: "Both dataframes were trained with the same training samples". In addition, we will include a plot comparing the DEM values and ground truth measured by a differential GPS which is going to be described in the Methodology section. The figure is shown as follows:*

[Figure]

*Where "DEM height" shows the elevation values from the Digital Elevation Model and "Height" is the elevation measured with the differential GPS in the field. Both have units of centimeters (cm).*

3. The introduction section needs to be reshaped to introduce the topic step by step. For example, you talked about remote sensing in line 20, and talk about wetlands, and then move to remote sensing again. Another example is that Lines 33- 40 are likely to appear in the study areas section.

*Thank you, we will clarify this part of the introduction*

4. Line 39. There are two references, i.e., Ward et al., 2016a and Ward et al., 2016. I did not find Ward et al., 2016a in your reference list. Another obvious error is Line 370. So I suggest the author please check this and also other references.

   *This is reviewed and clarified.*

5. Line 70. Please define VI, although you have defined it in the abstract. In addition, there are too many abbreviations, some of which were just used a few times, making the manuscript difficult to read and understand. So please remove unuseful abbreviations. And I also suggest the authors construct a table to explain each abbreviation.

   *We will add the list of abbreviations to clarify them.*

6. Line 90 - 95. Please specify the manufacture of your drone here. Please also specify the procedure of your radiometric correction, parameters and models used here. I think they are also useful to other researchers to do similar things.

   *The manufacturer is senseFly and they do not provide the internal functioning of radiometric correction. We follow the best practices from senseFly to carry out the radriometric correction by using the Airnov panels.*

7. Line 95 – 100. Can you also please show the confusion matrix for your classifications here? This would help to show that your selection of RF makes more sense.

   *It is possible to find in the referenced article (Prentice et al. 2021) the metrics of accuracy, kappa and comparison metrics between the two classifiers that the authors used in that article. The present study does not aim to show all the details of the referenced one but we will show the exact numbers of metrics.*

8. Line 123. Can you please explain more about the accuracy of the DEM used here. From my point of view, the accuracy of lidar-dem over wetlands is a bit low. So I think the specification is useful to show the robustness of your method.

   *We will provide the RMSE values between ground-truth DEM values and DEM model used in this section with the figure mentioned in point number 2 of this document.*

9. Table 2. I am not sure I lost something. But I do not know what @ means in this table. Please explain it in the table caption. And please show the unit (probably nm) of each band.

*We will add the term "nm" for nanometers. The symbol @ means the bandwith and we will specify it too.*

10. I think the first row of each table can be highlighted, making it easier to read and review.

    *We will do this change.*

11. I also suggest that Figs. A1 and A2 can appear in the main text to better display the accuracy.

    *We considered including these figures in the main text before but as we have already eight figures and four tables, we keep them in the annexes.*

12. In the discussion part, I suggest the authors explain the value of the proposed method in future research.

    *This part will be clarified and extended in order to explain more about the impact of this study.*

*Again, thank you for the comments. I hope we have solve them. We will work to make these changes in the main manuscript.*

---

## Author Comment (AC2)

*We thank the reviewer for their effort and constructive comments received. The key issues identified are similar to the first reviewer's comments and we are addressing them in the final submission of the article. Any additional figures to include, tables or references will be done in the main text too or appendices.*

*However, we answer to the comments and suggestions in this document.*

*Italic font type: Author's replies.*

In this paper the authors use a Random Forest model to derive the Plant Fractional Cover by upscaling UAV multispectral data to Sentinel-2 MSI data. They present a method that allows to overcome the limitations of low-resolution images from satellite data by using UAV. The topic fits the purpose of Biogeosciences and the special issue. However, I believe the manuscript needs to go through several revisions as I have comments and questions to the authors regarding the methods, the results, and the overall presentation of the study.

TITLE: Here you use the Plant Community Cover term but in the paper the Plant Community Fraction is used. I suggest correcting that.

*This comment suggests a different title and we agree with it. If the Editor accepts it too, we will changing it as "Plant Fraction Cover".*

INTRODUCTION: the authors provided a nice introduction to the topic of the paper. They provide information about the advantages of using remote sensing to monitor wetlands, pros and cons of using UAV and Satellite-based imagery, and the potential in the upscaling of UAV images to satellite resolutions. However, the introduction does not clearly state what is the novelty of the study. From my understanding of the last paragraphs, the implementation of Machine Learning (ML) models and especially Random Forest (RF) models to infer Plant Fractional Cover (PFC) is not new. At the same time it is not new to couple these models with DEMs as ancillary data. The last sentences suggest that this is what the study aims to. I think the introduction need to point out what are the new aspects the authors are looking at. What does this paper add to the current state of the field? Was sentinel-2 never used? Is the scale of the study new?

*Thank you for your comment.*

*This study provides the novelty of testing an upscaling methodology from fine-scale images taken with drones to a broader scale at Sentinel-2 image spatial resolution over Baltic coastal meadows to model the plant fraction cover of five plant communities. Several studies have*

*successfully monitor plant communities using high-resolution images with Unmanned Aerial Vehicles (UAVs) (i.e. Villoslada et al. 2020). However, to overcome the limitations of using UAVs, we studied the possibility of monitoring the plant communities with Sentinel-2 using UAV as reference. Up to this point, no one has been implementing this upscaling process in Baltic coastal meadows of Estonia.*

I also fell like the authors could reorganize a bit the structure of the introduction. At the beginning you mention remote sensing, then move to wetlands, and then back to remote sensing. I would first introduce the wetlands and then the remote sensing aspect. It makes the reading more fluid.

*Thank you; we will clarify this part of the Introduction, rearranging its structure.*

LINE 39: There is a Ward et al. (2016) and Ward et al. (2016a) cited but only one referenced in the bibliography. Is it a typo? Please correct or add the citation to the list.

*Thank you, this has been reviewed and corrected.*

LINE 71: Acronym VI was not introduced before.

*Thank you, this has been reviewed and corrected.*

LINE 75: What's the tidal range here? Do these areas go underwater regularly at each tidal cycle? During high tide/spring tide? Or only rarely during storms?

*Thank you for this question. The tidal range varies mainly due to storms. The answer to this question is going to be added in the main text and we answer it in the following lines:*

*The tidal variation in coastal wetlands is approximately 2 centimeters characterized by irregular inundations (floods) dependent on fluctuations of meteorological conditions across the North Atlantic and Fennoscandia (Kont et al., 2003 and Ward et al., 2016).*

*In addition, a table with a description of plant communities (as suggested in this document) will explain the different inundation regimes affecting them. The areas can be affected by floodings, affecting the plant community distribution.*

LINE 76: Here you use Figure 1 to reference to the figure. Many times throughout the paper you use the short form Fig. to reference to figures. I suggest picking one style and being consistent.

*Thank you, this format error has been corrected.*

LINE 82: I have to say that I am not familiar with the plant species here considered. I think it would be important to provide more information on these species/communities. For instance, what is their phenology? This would be a very important information. If they have a growing season in a specific time window, it means there is a limited period of the year of useful remote sensing images.

*We have added a table providing the information requested as found in references*.

FIGURE 1: This is just a personal preference a very subjective suggestion: I would increase the font size, especially for the latitude and longitude coordinates. I would also identify the study sites not with numbers but with their names and acronyms directly in the map instead of giving a legend in the caption. It would be much easier to locate them.

*Thank you for this suggestion. We have increased font sizes in all the figures. This improves the readability.*

LINE 96: Would it be possible to provide the confusion matrix for the classification. It is the best way for the reader to quickly assess the quality of the classification. You could simply put it in the Appendix.

*We have added a new figure in the appendix , which includes the confusion matrices per study area corresponding to the number of pixels in the test fraction classified per each plant community. These derive from the study of Prentice et al., 2021.*

[Figure]

*Results from the Random Forest pixel classification. MAT (Matsalu), KUD (Kudani), TAS (Tahu South), RAL (Rälby), TAN (Tahu North) and Rumpo (RMP). Kappa values are MAT: 0.98, KUD: 0.92, TAS: 0.93, RAL: 0.89, TAN: 0.99 and RMP: 0.99.  Each class of Predicted and Actual Plant Communities are LS (Lower Shore), OP (Open Pioneer), US (Upper Shore), TG (Tall Grassland) and RS (Reed Swamp).*

LINE 101: I understand that the classification is based on a different study. However, since it is key to the analysis, the authors should provide a few more details on the classifications (see comment above).

*Thank you. We have added the confusion matrices (shown in the previous comment) and also the dGPS used to measure the ground truth height necessary to validate the Digital Elevation Models. Hyperparameter tuning of Random Forest will be also provided. Table A1 shows the presence of plant communities surveyed in each study area.*

LINE 107: The analysis used only one Sentinel-2 scene. Do the authors think that the study would benefit from using multiple scenes? Since the authors say that the method has implications on monitoring (i.e. using multiple images), I think that testing the model on a different scene would prove that the method is more robust. Having said that, I am not suggesting that the acceptance of the paper should depend on this additional analysis as I believe it is a serious task to undertake.

*This is a very interesting suggestion. Indeed, we consider contributing to the study of Baltic Coastal meadows with a  time series of Sentinel-2 in the visible – near infrared spectrum. The present study is a first approach to estimate the PFC using Sentinel-2 images having the centimeter-resolution UAV images as a reference. Further work will be focused on monitoring Baltic Coastal meadows in a broader period.*

Was the Sentinel scene taken during low tide? I think the presence of water can affect the reflectance. The authors should specify (maybe in the discussion) what kind of images are good for this method. In the discussion there is only the mention to take UAV images close to the satellite passage. The authors can expand here on what are other important aspects that goes into the choice of the scenes.

*Thank you for this suggestion. The presence of water affects the spectral response of plant communities but not during the dates chosen. However, the drone flights were carried out during high phenological activity and when the weather conditions were optimal for those flight plans. Including the Red-Edge band enhances the plant community detection with images as this band is more sensitive to chlorophyll reflectance. This is why we chose Sentinel-2, too.*

*We consider this comment for the discussion and further work related to the previous comment: Studying the affection of presence of water to plant community detection from a time-series approach.*

TABLE 1: I think Table 1 is missing some information. I see only the study area and the drone flight dates, but no tile number or satellite overpass date is reported. Either reformulate the caption or add the information in the table.

*Thank you, the table caption is corrected. This table includes the dates of drone flights. The main text mentions the tile number and date of the Sentinel-2 image acquisition because it was the closest to the drone flights.*

TABLE 2: units of wavelength and spatial resolution are missing.

*Thank you, these have been added in units of nanometers (nm)*

LINE 122: could the authors provide the vertical error of the DEM? Since the microtopography is important, it would be useful to know what's the vertical error.

*This has been added as a new figure in the main text. Please see below the figure:*

[Figure]

*The dGPS height is the height measured with Sokkia GSR2700 ISX differential global positioning system (dGPS). The total RMSE is 0.06 cm.*

FIGURE 2: Like for Figure 1, I suggest to directly use site acronyms in the map. I also suggest increasing the size of the scale bar. It's very difficult to read it.

*Thank you, the scale bars have been modified for a better readability. We have replaced the numbers with the acronyms of the study areas.*

LINE 137: 'A correlation and a linear function were used' is not very precise. From this I understand that there are two levels of evaluation. What kind of correlation? Is the correlation found with a linear function? Please rephrase for more clarity.

*Thank you, we will rephrase this statement in the main test, as follows:*

*The comparability and consistency of the spectral data from PS and MSI bands was analyzed by fitting the values in a linear model, calculating the coefficient of determination ($R^2$) and Root mean squared error (RMSE). The p-value showing the significance of the relation between PS and MSI.*

LINE 138: Does the averaging of the elevation smooth the microtopography? Do you think this step has an effect on the performance of the model?

*Thank you for this question. Different aggregation methods produce different results on the distribution of values within the spatial unit of a pixel considered here, affecting the spatial characteristics too.*
*We will include an explanation in the main text. As follows:*
*We considered that aggregating with the average value in a spatial unit of the pixel produces more predictable behavior (Bian and Butter, 1999).*
*The purpose of this study was not the comparison of different aggregation methods but we agree that using different aggregation might affect the performance of the predictions.*

LINE 140: I am a bit confused on the separation of DF1 and DF2. DF0 is a sub-sampled dataframe from DI, and it already contains the elevation variable since the DEM information was previously added. Thus, I understand that DF2=DF0, and it is not really a new dataframe. Would it be easier to consider just two dataframe? One with and one without elevation?

*After undersampling the initial dataframe (DI), we obtained DF0. We decided to follow a sequential order, so the following step was to split this dataframe into two new and different ones (DF1 and DF2). DF1 is used to predict without the variable of Digital Elevation Model and DF2 is used to predict with that variable.*

LINE 155: I think the reference to Figure 1 is wrong. Please check.

*Thank you, this has been corrected.*

LINE 169: I assume that in each MSI pixel we want to have the sum of all PFCs equals to 1. If you have a separate model for each plant community, how do you make sure of that? Do you force it somehow? Please clarify this as I think is an important step.

*Thank you for this comment. We can clarify this in the main text.*

*The initial dataframe (DI) contains all the individual PFC in different columns. In the figure caption of figure 3 of the preprint version , "yi" (where " i " can be LS, OP, US, TG or RS) is the PFC of each plant community and their sum  is 1 (100% cover). Then, each "$y_i$" is used for one single model. For example, in the Random Forest model for LS, we train with the PFC of LS and then test it.*

LINE 173-174: I am not sure the reference to Figure 4 is correct: it looks more correct to reference these sentences to Figure 5.

*Thank you, we have corrected this.*

LINE 174: rewrite as 'predefined hyperparameters'.

*Thank you, we have corrected this.*

RESULTS: Here you clearly show that elevation is a key feature to predict vegetation zonation. That makes 100% sense. I think it would be important to show the DEM to the reader due to the importance of this parameter. That would help to understand whether a species prefers a high or low area, which is directly linked to the ability to withstand a low or high hydroperiod.

*This will be included in the table suggested before to describe the plant communities under study. This table will summarize this comment as well. In addition, an average height range will be provided.*

LINE 185: it would be good to add a figure were you show the correlations between the MSI and PS reflectance in the MSI GRID. Maybe this figure can be shown in the Appendix/Supplementary Material.

*We have added this figure in the appendix section, for each spectral band. Please, see below:*

[Figure]

*This figure clarifies the R², RMSE and p-value obtained from the linear fitting between bands. X and y axes are in reflectance units (%) as well as RMSE. Correlations in all the cases are significant,*

*We apologise because the previous numbers in the preprint version were wrong. As seen in this figure, R² and RMSE are different as in the preprint version. Therefore, we will update it and review the Discussion.*

LINE 194: I invite the authors to consider moving Figures A1 to A4 to the main text, since they show the goodness of the models. Maybe Figure 2 can be moved to the Appendix to avoid overloading the main text with figures. It just shows the grid, so it is not as useful as the other 4 figures.

*Thank you, we have moved the Figure 2 to the appendix, as the comment suggests, it is not as informative as the Figures A1 to A4.*

TABLE 5: It would be better to specify the p-value instead of simply indicate <0.05.

*The p-values are very low, under 0.0001, then, we will specify this number instead of 0.05.*

LINE 216: I would be more specific. Are your results comparable to those studies you mention? I think it would be better to expand here the discussion and show a

comparison with other similar studies. It would be very informative to know what you did better. Did these studies consider DEM or only VIs?

*Thank you for this comment. The studies mentioned in this line also have found positive correlation between bands for an upscaling methods, although they did not model plant community distribution. Padró et al. 2018 used an exhaustive comparison between sensors, two of them were MicaSesnse and Sentinel-2 MSI, finding good correlations. Díaz-Delgado et al. 2019 used the same camera than in this study, Parrot Sequoia, on board of a fixed-wing drone and compared its spectral bands with Sentinel-2 MSI. Both studies are done in the Iberian Peninsula, whereas our study is done in the Baltic region, agreeing with a good correspondence between a multispectral camera on board of a UAV and the MSI of Sentinel-2.*

LINES 210-224: I think this part is not really useful. Here you are just repeating the methods and giving more results. You can move lines 222-224 in the results sections.

*Thank you for this comment. We found good correspondences between spectral bands of Parrot Sequoia and MultiSpectral Instrument (MSI in Sentinel-2), similarly to other studies mentioned. We consider that this is an important result to discuss in spite of having different dates between the UAV flights and  the Sentinel-2 overpass. But we might considered the rest of the lines you mention to be included in the Results section.*

LINE 235: 'figures 6 and A1' should be 'Figures'.

*Thank you for this comment. We have corrected this.*

LINE 236: Typo metre. Please correct. Also when you use the number use the unit. So in this case it should be '1 m'. Please make sure to follow that throughout the paper.

*Thank you for this comment. We have corrected this.*

LINE 237: Are you sure that the reference to Figure A2 is correct? I think the correct reference is Figure A3.

*Thank you for this comment. We have corrected this.*

LINE 238: Looking very quickly at Figure A4, elevation seems to be the most important by far with the only exception for the OP class. If I read correctly for the other classes, elevation in terms of importance it is around 0.5, and 3-6 times more important than the VI with highest importance. One could argue that acceptable results could be achieved with only elevation. Have you tried to do that? Can you comment on this?

*Thank you for these questions. We used the Digital Elevation Model as the only independent variable in Random Forest Regressor model to predict the Plant Fractional Cover but its results had higher RMSE than the rest of the models with a very low variance explained. We also trained using different hyperparameters but without good results. This is because the Vegetation Indices calculated from the spectral bands are also indicators of the presence of vegetation, their phenological activity and their density because bare soil specially modifies the reflectance in the Red Edge and Near Infrared bands of the spectrum. We include this comment in the discussion.*

LINE 239: This comment concerns the entire manuscript. I had hard time to remember all acronyms. I know that they make the writing faster, however I think it would be better to reduce the number. Maybe the authors can simply pick the most used ones.

*Thank you, as another suggestion, we are including a table of abbreviations in the Introduction section.*

LINE 242: I think the authors could expand here. Why these communities are so dependent on elevation?

*The elevation (microtopography) determines the salinity, flooding periods, nutrient fluxes and topsoil moisture. This is explained in detail in Ward et al., 2016. Moreover, we will include this clarification in the main text.*

LINE 245 to 248: The sentence starting with 'Overall, …' is badly written. I suggest reformulating and breaking it into short sentences.

*Thank you, this is corrected.*

LINE 251: Do the authors think that other satellite data would have been applicable to the study (e.g. Landsat)?

*Not for this study. We used Sentinel-2 for some reasons: the Red-Edge band (band 6 of MSI sensor), its high spatial resolution (10 meters and, as mentioned in the methodology section, downscaling the Red Edge band to 10 meters using the Superresolution Algorithm in SNAP software), the high temporal resolution (5 days reached by the Sentinel-2 constellation, thus, more chances to get an image closer to the flight dates with low cloud coverage) and its public and instant availability. On the other hand, Landsat does not have Red-Edge band in its sensors.. In addition, the spatial resolution is higher.*

*As mentioned in a previous comment, a multi-sensor approach, the fusion Sentinel-2 and Sentinel-1 might improve the PFC prediction models.*

LINE 257: Can the authors suggest other ancillary data for future research besides DEM? Maybe inundation time?

*Yes, and this will be added to the main text. The use of a multi-source sensing approach has been done for these wetlands, where the use of Synthetic Aperture Radar (SAR) together with optical images from UAV reveal inundation patterns by sporadic seasonal storm surges. Thus, the SAR sensor is an optimal candidate but for a fusion with Sentinel-2 images. On the other hand, we can suggest the use of aspect, grazing management history and distance from the coast as ancillary data because, in spite of the low tidal ranges, the study areas do not have similar effects due to open water.*

APPENDIX: The axis ticks are small and hard to read. Especially Figures A3 and A4. In Figures A1 and A2 you are not showing the predicted error. You are just comparing predicted PFC with measured PFC. When you are comparing modelled and observed values you do not need to compute the $R^2$ since you are not really looking for a model between the two. The RMSE values is a good index to evaluate the goodness of your predictions. Maybe you could add a second index like Model Efficiency or Percentage Bias (it doesn't necessarily have to be these ones). Especially with the second one you could quantify the general tendency of your model to underestimate, and overestimate observed values.

*Thank you for this comment. We keep the metrics and will include the Mean Bias Error in these figures as well as in the methodology.*

What is the unit/variable in y axis in Figure A3 and A4? Is it the explained variance for each variable? Please clarify.

*Thank you for this question. We will clarify this in the text, adding "Variable Importance", clarifying in the figure caption that this variable importance ranges from 0 to 1, indicating the contribution of each single feature (variable) to each of the tree's total impurity reduction. In Random Forest, the importance is calculated as the average of importance over all trees.*

---

## Author Comment (AC4)

RESULTS: Here you clearly show that elevation is a key feature to predict vegetation zonation. That makes 100% sense. I think it would be important to show the DEM to the reader due to the importance of this parameter. That would help to understand whether a species prefers a high or low area, which is directly linked to the ability to withstand a low or high hydroperiod.

*Instead of including this suggestion in the table mentioned in the previous reply, we are showing the distribution of plant communities in relation to the elevation in the following figure, as it describes how the plant communities vary according to the elevation in each study area. Kudani (KUD), Matsalu (MAT), Rumpo (RMP), Rälby (RAL), Tahu North (TAN), Tahu South (TAS).*

[Figure]

*Plant communities: Lower Shore (LS), Open Pioneer (OP), Upper Shore (US), Tall Grassland (TG), Reed Swamp (RS). This figure is going to be included in the annex.*

---

## Referee Report (RR1)

The authors have taken into consideration all my comments and those from the second reviewer. They have improved the quality of the presentation of the work with more precise writing and figures. I suggest the acceptance of the paper. I have just a couple of suggestion and points to note (line number refers to the version 2 of the manuscript):

LINE 88: I would simply say "… in relation to elevation in a boxplot…"

LINE 106: typo elevation7

FIGURE 2: RMSE in figure does not have the unit

---

## Author Response (AR2)

**Answers to reviewer 1**

Thank you very much for these concise reviews and suggestions. We have considered all of them carefully while reading the manuscript again. We answer to the comments and suggestions in this document.

*Italic font type: Author's replies.*

My main concern, how the authors rescale the random forest predicted PFC is still there, although there is a sentence in 'All the grids sum a total of 1 (100%)' line 169. But if I understand correctly, the authors used independent RF model to predict the PFC of each community (Figure 12), how the authors to rescale these PFCs.

I mean, can values in each pixel in Fig 12 sum to one? If so, please highlight it to make clear to the readers. I appreciate your effort here, it is very important here. I also suggest the authors to display the accuracy before and after the rescaling (maybe in supporting information, please see the first two rows in Fig. 9 in Immitzer et al. (2018), doi: 10.1016/j.rse.2017.09.031 and Figs. 7e and f in Yang et al. (2020), doi:10.3390/rs12193224) to highlight the importance of this effort.

*Many thanks for this useful suggestion. Indeed, the previous text in line 169 was somewhat confusing. We have now addressed this issue by rescaling the predicted PFC values to a range of 0-100 per plant community, ensuring that the total sum of predicted PFC within a MSI pixel equals 100. Subsequently, these rescaled predictions are validated with the same validation datasets used in the non-rescaled predictions to ensure comparability. The construction of this test set is explained in the Methodology and referred to Figure 5. We have added a new figure in the main text of the manuscript to visualize these results, as suggested. The methodology for this additional analysis is described in the Methodology section (lines 192-194), while the new results are presented in the Results section (lines 225-228), and a description of the implications of using this validation with rescaled PFC in the Discussion section (lines 285 – 292). Now, the Fig 12 you mention is Fig 13.*

As for the feature importance analyses, I think it makes sense until the author highlight the high accuracy after the rescaling.

*Here we would like to point out that feature importances are generated during the training of each RF regressor model with the training datasets from DF1 or DF2. The rescaling itself is carried only on the raster layers resulting from the RF regressor. This means that the feature importance analysis is in fact not affected by the rescaling and therefore, the importances remain the same as before the rescaling.*

I also suggest the authors can read the manuscript more and take care of some worlds and grammars. There are some errors. There are some minor concerns:

1. For the title, I suggest that it could be the synergistic use of UAV and DEM to map fractional cover in Sentinel-2 pixel?

*We appreciate this suggestion; however, we have opted to not include this modification in the title. We based our choice of title on publications such as Emilien et al. (2021) (see in the Reference section), which*

*clearly refer to the synergistic use of satellite-level data and UAV-level data. The DEM is just one more ancillary dataset in our study.*

2. Line 1: threats?

*This has been corrected, thank you.*

3. Line 2: monitoring and assessment what? And are, instead of is.

*We have changed this line to : Thus, their monitoring (coastal wetlands) and assessment is vital for evaluating their status, extent and distribution*

4. Line 16: I think climate is a kind of environmental factors.

*We changed this, however, see the comment of suggestion 6.*

5. Line 18: I do not think it is necessary to use the ESs. I suggest the full spelling is more readable.

*We changed this, however, see the comment of suggestion 6.*

6. Lines 12 – 31: I still suggest there still need some improvements. You probably can go to coastal wetland is important/how and go to function of vegetation. And then go to the importance to monitor vegetation communities.

*Our introduction starts with the importance of coastal wetlands with relevant references carried out on coastal wetlands, followed by an introduction of Baltic Coastal Meadows and the use of Remote Sensing to monitor the distribution of plant communities in these ecosystems.*

7. Line 58: are essential?

*This has been corrected, thank you.*

8. Line 61: Climate changes?

*This has been corrected, thank you.*

9. Line 90: characteristic, I suppose typical may be more precise.

*We replaced "characteristic" by  the term "typical". Thank you.*

10. Line 116: elevation7?

*This typo has been corrected. Thank you.*

11. Line 129: Band 6 of MSI.

*The preposition has been changed. Thank you*

12. Lin4 188: VIs?

*This has been corrected. Thank you*

13. Table 5: there is a '12' behind MGRVI, what does it mean?

*This was a typo. Thank you.*

14. Line 297: et al. (2021)?

*That is right. It has been corrected, thank you.*

**Answers to reviewer 2**

We appreciate your considerations. Thank you very much for these last reviews. We have considered all of them carefully while reading the manuscript again.

*Italic font type: Author's replies.*

LINE 88: I would simply say "… in relation to elevation in a boxplot…"
*Thank you, this suggestion clarifies this sentence. We have changed it.*

LINE 106: typo elevation7
*Thank you, we have corrected this typo.*

FIGURE 2: RMSE in figure does not have the unit

*Thank you, this has been corrected.*

---

## Author Response (AR3)

**Answers to reviewer**

Thank you for your concise reviews and valuable suggestions. We have carefully considered each of them while revisiting the manuscript. Our responses to the comments and suggestions are provided in the following lines.
*Italic font type: Author's replies.*

The authors test the possibility to estimate the relative cover of different coastal communities by coupling UAV image and public-available but coarse image. It is interesting for me and of high importance for future coastal monitoring. And I appreciate the authors try to solve my concerns. I think it is good to product if the following concerns solved.

My main concern is, for me, Fig. 8 and Fig. 10 looks very similar. I suppose the authors may use them mistakenly. So I suggest the authors please double check and correct them if it is needed. After that, I suggest that the Fig. 10 can go to supporting information.

*Thank you very much for this comment, we have fixed this issue. Figure 10 shows the lower performance of RF regressions trained with DF2 after rescaling the predicted PFC. Figure 8 shows the prediction errors of models trained with DF1.*

There are some minor issues from my point of view.

1) Line 7: I suppose Vegetation Indices (VIs) makes more sense than vegetation indices (VI) here. And all 'VI' over the main test may be plural.

*Indeed, it is easier for the reader to use the abbreviation VI for Vegetation Index, and then, use VIs for the plural form. Accordingly, we have modified this throughout the text and in the Table 1.*

2) Line 227: which models were not used map? I did not understand. Can you please make it clearer?

*The models that were not used to map the PFC over the study areas were those trained with DF1 and rescaled values from DF2. Thus, only models with relative values of PFC from DF2 were used to map the PFC over the study areas. We have clarified this point and simplified the lines 224 to 231.*

3) Lines 254-257: I agree with the fact that the mixture is one of the reasons for uncertainties of RF. But do you think it also can be attributed to the nature of the RF. The RF regression makes prediction by using the average value of each tree inside, thus avoiding the appearance of extremely low or high values. Do you think this can be solved by including dominant and mixed samples? It is just my speculation. What do you think about that?

*In Random Forest modeling, dealing with uncertainty and ensuring the model is interpretable is a challenging task. Indeed, the averaging of individual tree predictions during training influences Random Forest regression predictions. We overcome the uncertainties with 1) sampling a balanced dataset that represents the entire spectrum of PFC values and 2) an approach to hyperparameter tuning with the Grid Search Cross Validation.*

*Before the under-sampling to balance the dataset, we studied the distribution of all the PFC values in the initial dataset (DI), having an unbalanced distribution towards the extreme values of PFC due to a*

*completely or partially absence of plant communities in a grid (S2 pixel) or high cover. We performed various Random Forest regressions keeping these values in order to test the stability of the model under different training scenarios. Models exhibited good scores during training but suboptimal test scores, suggesting the possibility of overfitting. Thus, we considered that keeping the unbalanced extreme values was not appropriate to train the models. In order to keep samples that are more mixed, we also tried reducing the interval of the bins and this led to worse performance of the models. The best approach was used in the present study.*

*We agree that RF may also incorporate a certain degree of uncertainty in its architecture. In our opinion, not only reducing the uncertainties may also depend on complexity and dataset characteristics but also the model complexity.*

*We have addressed this discussion in lines 250 -259.*

4) Line 173-174: how do you carry out the under-sampling? Just randomly select some? Can you please specify here? One question from curiosity, do you think the under-sampling emphasize the above-mentioned over/under-estimation problem?

*The under-sampling was done by selecting the S2-pixels randomly with PFC values contained in a bin. The number of S2-pixels per bin equals the number of values in the minority bin.*

*The under-sampling strategy was done to balance the initial dataset (DI) to avoid the model to learn from one bin more than another, avoiding potential overfitting. However, we appreciate the question from the reviewer and we think that over or under-estimation could be improved by constructing an exhaustive training dataset from field survey plots equal to the area of S2 pixels. However, this is high time consuming due to logistic issues.*

*This clarification is added in lines 169 to 175 and discussed in lines 255-259, because they align with the previous comment.*

5) Line269: As for the importance analyses. I am curious about that VIs represent spectral information, or different fusions of spectral bands. So different VIs, although calculated via different equations, contain somehow similar information. So I think it is a bit arbitrary to say DEM is the most important variable for some cases, although importance value from RF is very high. However, if we consider all VIs are spectral information, your analyses show that spectral information is more important than elevation information in all cases. How about your thoughts about that?

*First of all, the Digital Elevation Model, although its coarse spatial resolution in this study, still represents the microtopography in these type of wetlands as seen in the results. Plant communities are strongly dependent on this variation.*

*In relation to the spectral information from Vegetation indices, the Random Forest algorithm evaluates the individual contribution of each Vegetation Index and Digital Elevation Model to a single tree's total impurity reduction, meaning that it calculates the importance independently. This is a relative measure of the model performance. The lack of additivity is due to the complex interactions and non-linear relationships considered by the Random Forest algorithm, which can lead to the different Vegetation Indices being more or less important independently of their calculation. In this way, the Digital Elevation*

*Model is relatively more important for the model to estimate the PFC. In other words, it makes more accurate predictions within the ensemble of decision trees that make up the Random Forest.*

*Since variable importance in RF is not an additive variable, we cannot not really claim that spectral information is more important than the DEM. In a similar manner, VIs are not additive representations of the electromagnetic spectrum. In that regard, we should not add them together and take it as "spectral information". We hope this answer is satisfactory and clarifies the idea of VIS and variable importance in RF.*

*In order to summarize all these points, we have added a clarification in the lines in Methodology section 189 − 191, and in the Discussion section in lines 280-284.*